Article 

# Dimensionality-dependent electronic and vibrational dynamics in low-dimensional organic-inorganic tin halides

Yanmei He [1], Xinyi Cai[2], Rafael B. Araujo [3], Yibo Wang[1,4], Sankaran Ramesh [1], Junsheng Chen [5], Muyi Zhang [2], Tomas Edvinsson [3], Feng Gao [2] ✉ & Tönu Pullerits [1] ✉

Photo-induced dynamics of electronic processes are driven by the coupling between electronic and nuclear degrees of freedom. Here, we construct one- and two-dimensional organic-inorganic tin halides to investigate how dimensionality controls exciton-phonon coupling and exciton self-trapping. The results show that a one-dimensional system has strong exciton-phonon coupling leading to excitation-independent self-trapped exciton emission, whereas a two-dimensional system exhibits over ten times weaker coupling resulting in free exciton emission. The difference originates from enhanced Anderson localization in a one-dimensional system. Femtosecond transient absorption experiments directly resolve room-temperature vibrational wave-packets in a one-dimensional system, some of which propagate along the self-trapped-exciton potential energy surface. A combination of wagging and asymmetric stretching motions (~106 cm$^{-1}$) in tin iodide is identified as such a mode, inducing exciton self-trapping. While no room-temperature wave-packets are observed in a two-dimensional system. These findings uncover the interplay between dimensionality-dependent exciton-phonon coupling and electronic/nuclear dynamics, offering constructive guidance to develop multifunctional organic-inorganic metal halides.

Solution-processed low-dimensional hybrid organic-inorganic metal halides have received considerable attention as promising materials for efficient optoelectronic devices[1–6]. Because of the soft ionic crystal structure, their electronic states can strongly couple to the lattice vibrations through the exciton-phonon coupling (EPC). This interaction tunes the optical and electronic properties of the material, opening relaxation channels for excitons and charge carriers[7–10]. For instance, in two-dimensional (2D) layered lead-halide perovskites (LHPs), pronounced EPC often leads to the formation of a large polaron, where the polarization of the local environment can shield

hot carriers, thereby slowing down the thermalization of the non-equilibrium photoexcited carrier population via electron/hole-longitudinal optical (LO) phonon scattering or Auger heating process[11–13]. Alternatively, since in ionic systems like LHPs, the Fröhlich coupling of the electrons and phonons engages mostly the low-momentum LO phonons, the hot carrier cooling selectively excites only those phonons, thereby rapidly establishing thermal equilibrium between the selected LO modes and the charge carriers. This can drastically slow down the further cooling of the hot carriers—an effect called hot phonon bottleneck[14,15]. The peculiar EPC thus inspires the

[1]Division of Chemical Physics and NanoLund, Lund University, Lund, Sweden. [2]Department of Physics, Chemistry, and Biology (IFM), Linköping University, Linköping, Sweden. [3]Department of Materials Science and Engineering – Solid State Physics, Uppsala University, Uppsala, Sweden. [4]Department of Nuclear Science and Technology, Nanjing University of Aeronautics and Astronautics, Nanjing, China. [5]Nano-Science Center & Department of Chemistry, University of Copenhagen, Copenhagen, Denmark. ✉e-mail: feng.gao@liu.se; tonu.pullerits@chemphys.lu.se

development of versatile LHPs for possible future efficient hot-carrier solar cells[16]. In one-dimensional (1D) and zero-dimensional (0D) organic-inorganic metal halides, EPC often results in self-trapped exciton (STE) emission with a large Stokes shift compared to the 2D counterparts. Usually, the STE emission has a high quantum efficiency, promoting its potential for developing white light-emitting diodes (WLEDs)[17–20].

The femtosecond pump-probe techniques, including impulsive stimulated Raman scattering (ISRS)[21], impulsive vibrational spectroscopy[22], ultrafast electron diffraction[23], and femtosecond transient absorption (fs-TA) spectroscopy[24], allow for the real-time observation of the photoexcited coherent phonon dynamics since a short laser pulse can excite a coherent phonon wavepacket. By using these methods, the carrier/exciton-phonon interactions in the 2D LHPs have been comprehensively explored. For example, the studies of coherent phonon dynamics in LHPs, including $(PEA)_2PbI_4$ and (FPT)$PbI_4$, reveal that the low-frequency vibrational modes (<60 cm$^{-1}$) involving the bending and stretching vibration in lead halide octahedra, play a crucial role in polaron formation and in that way modify the hot carrier cooling process[12,22,24,25]. As reduced dimensionality enhances both quantum confinement and lattice fluctuations, the strength of EPC is inherently sensitive to a material's dimensionality. Although strong EPC has been widely reported in low-dimensional halide perovskites, these effects are usually discussed in terms of enhanced confinement or polaron formation, rather than in connection with dimensionality-induced exciton localization. In particular, the role of such localization in governing EPC remains largely unexplored for organic–inorganic tin halides, where dynamic disorder and soft lattice modes are especially pronounced. Understanding how dimensionality and localization jointly determine EPC provides a new perspective on carrier–lattice interactions and offers practical routes to tailor the optical and electronic properties of lead-free tin halides for environmentally sustainable applications in solar cells, light-emitting diodes, and coherent light sources.

In this work, we tune the ligand concentration to prepare the 2D and 1D Dion-Jacobson (DJ)-type organic-inorganic tin halides dressed by a soft ionic lattice, leading to distinctly different coupling strengths between the lattice vibrations and electronic structure in these two systems. As we will show, the dimensionality has consequences for the properties of excitons and their dynamics – in a 2D system, free excitons (FEs) dominate, while in a 1D structure, STEs are formed. Using fs-TA spectroscopy under 400 nm excitation, we reveal intrinsic differences between the FEs and STEs by comparing their excitation-dependent electronic and temperature-dependent vibrational dynamics. The STE population dynamics are independent of excitation intensity up to very high exciton densities (>$2.9 \times 10^{20}$ excitation/cm$^3$/pulse), while in most 2D perovskites, Auger recombination appears above $1.0 \times 10^{18}$ excitation/cm$^3$/pulse. Moreover, coherent oscillations are observed in the 1D system at room temperature, while the coherence is absent in the 2D system. Based on Fourier transform analysis and theoretical modeling of oscillatory wavepacket dynamics, we establish a microscopic picture of vibrational dynamics in 2D and 1D systems. We conclude that the ligand-modified EPC significantly influences the phonon anharmonicity and lattice distortions in the tin iodide, also altering the electronic dynamics. These studies unravel the underlying mechanism for the dimensionality-dependent emission and shed light on the lattice dynamics in low-dimensional organic-inorganic tin halides.

## Results

### Structural and photophysical properties

DJ-type organic-inorganic tin halides $ODASnI_4$ and $ODASn_2I_6$ (ODA = 1,8-octane-diamine) thin films were prepared by adjusting the molar ratios between the organic cations and the tin halides (Supplementary Note 1, Fig. 1a, b)[26,27]. The morphology, elemental composition, and photoluminescence quantum yields (PLQYs) of these materials are shown in Supplementary Figs. 1–3, demonstrating their good quality. The powder X-ray diffraction pattern of $ODASnI_4$ thin film presents periodic peaks at 13°, 20°, 27°, and 34°, which are similar to those in 2D $ODAPbI_4$, while the periodicity is absent in $ODASn_2I_6$ thin film, accompanied by the appearance of a new peak at 8° (Supplementary Fig. 4)[28]. Fig. 1c displays the narrow emission of $ODASnI_4$ thin film and the broad emission of $ODASn_2I_6$ thin film. Given the distinct emission features and diffraction patterns, we thus conclude that the 2D phase is dominant in $ODASnI_4$ thin film, while the $ODASn_2I_6$ thin film has a 1D nature. The real structures are probably disordered to some extent, but the main feature, either being 1D or 2D, dominates these two systems. In the following text, we refer to them as 1D and 2D systems. In the absorption spectrum, we observed the excitonic peaks at 502 and 588 nm for the 2D system and 359 nm for the 1D system (Fig. 1d). The low-energy peak (588 nm) in the 2D system corresponds to the lowest exciton transition 1 s. The 502 nm band originates from the higher exciton transitions together with the band edge absorption. The PL spectrum of the 2D system shows a relatively small Stokes shift of ~27 nm ($\lambda_{em}$ = 616 nm) with a weak low-energy tail. However, the 1D system has a significantly larger Stokes shift of ~251 nm ($\lambda_{em}$ = 610 nm). We also notice a substantially longer PL lifetime and higher activation energy in the 1D system compared to the 2D one (Supplementary Figs. 5–7). These distinct photophysical properties suggest that the electronic transitions in 2D $ODASnI_4$ and 1D $ODASn_2I_6$ are dominated by FE and STE states, respectively.

As the interplay between the electronic state and nuclear motion is key to shaping both optical properties and dynamics, we aim for a quantitative description of the exciton-phonon interaction strength. For that, we use the Huang-Rhys ($S$) factors of the vibrational modes that we determine both via experiments and computations. First, we analyze the temperature-dependent fluorescence line width as $FWHM(T) = 2.36\hbar\omega\sqrt{S \cdot (2n_T + 1)}$ where FWHM is the full width at half maximum, $n_T$ is Bose-Einstein occupation number, and $\hbar\omega$ is the effective mode energy (Fig. 1e, f)[29,30]. The obtained $S$ factor for the 1D system is ~36 with the effective mode frequency of 160 cm$^{-1}$. We also analyze the bandgap changes as a function of time via a combined molecular dynamics – electronic structure calculation. From the Fourier transform of the bandgap fluctuation autocorrelation function, we obtained a spectral density function that provides mode frequencies modulating the bandgap and the relative strengths of the corresponding $S$ factors, as shown in Fig. 1g–i and Supplementary Fig. 8. The final $S$ factors were derived from an analogous expression as above where instead of fluorescence FWHM, the root mean square of the bandgap fluctuations was used (for more details see Supplementary Note 2)[31–33]. The fluorescence linewidth analyses lead to a larger reorganization energy $\lambda = \sum_i S_i \hbar\omega_i$ than the calculated bandgap fluctuations because the higher energy tail of the spectral density function stretches beyond what is represented by the 3 modes that are used in the analyses. The general agreement of the two methods provides evidence for a strong displacement of the excited state nuclear equilibrium position compared to the ground state. This leads to the formation of the STE state and distorted lattice at the excited state of the 1D system, causing the broad red-shifted emission. In the 2D system, the overlap with the absorption band might lead to partial self-absorption, which could result in an underestimation of the FWHM. Additionally, the weak emission tail at 660 nm, coming from the surface trap states (SUT), makes the line-width analysis for the $S$ factor unreliable (Supplementary Fig. 9). Below, we will quantify the EPC strength of the 2D $ODASnI_4$ using the method derived from the oscillatory components in the fs-TA measurements.

### Electronic dynamics

TA measurements used excitation at 400 nm (~3.1 eV) and a broad white light probe, allowing for to collection of detailed information on

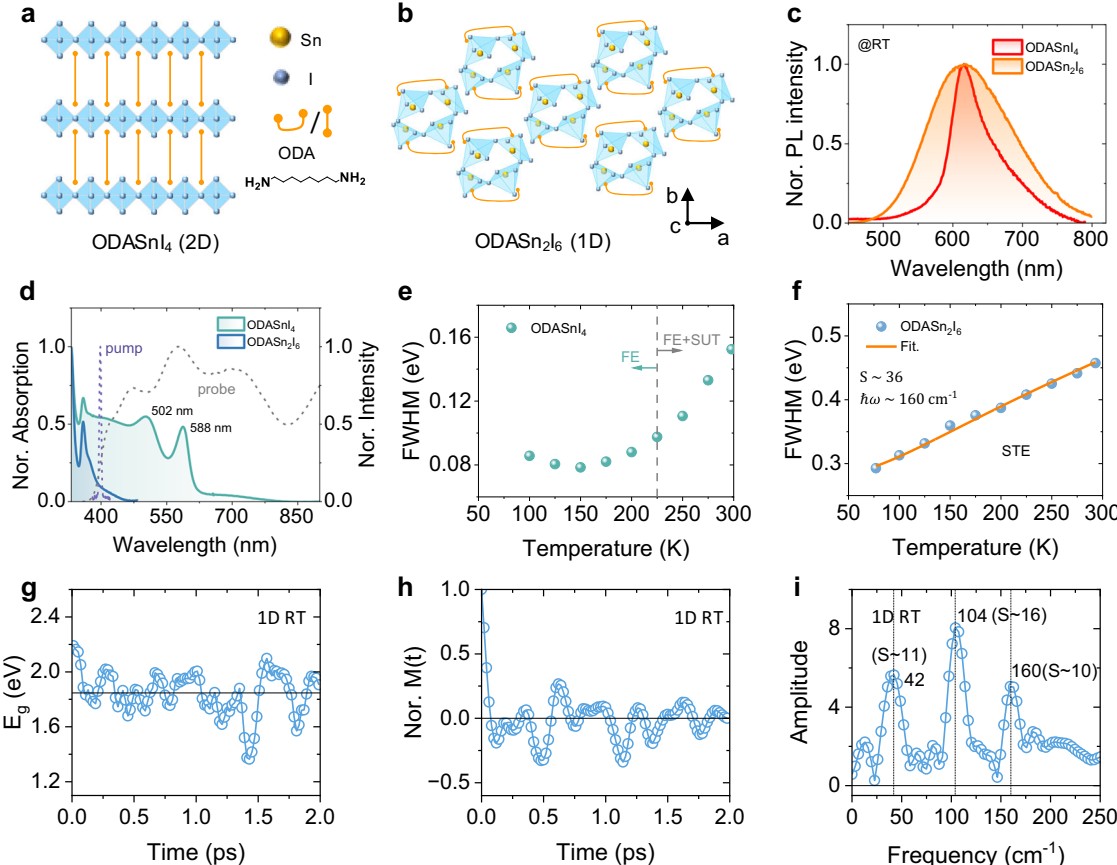

**Fig. 1 | Photophysical characterization and exciton-phonon coupling estimation.** 2D ODASnI₄ and 1D ODASn₂I₆ thin films: **a** Normalized UV-vis absorption spectra. Normalized pump, and probe spectra in fs-TA measurements; **b** and **e** Schematic crystal structure (In the 1D structure, the four corner-sharing [SnI₅]³⁻ moieties align along the c-axis like a glide plane separated by ligands)[27,28,73]; **c** and **f** Temperature dependence of FWHM; **d** Normalized PL spectra; For the 1D system: **g** Calculated energy gap time-correlation function; **h** Normalized autocorrelation function $M(t)$ of the energy gap; **i** Fourier transform spectrum of $M(t)$. The Huang-Rhys factor and effective mode frequency are shown for comparison.

exciton dynamics. As shown in Fig. 2a–f, the 1D and 2D systems show very different TA signals. In the 2D system, we identify two ground-state bleaching (GSB) bands at 506 and 584 nm, which are consistent with the excitonic peaks observed in the steady-state absorption spectrum. Three excited-state absorption (ESA) bands are observed together with the two GSB bands. The ESA bands can have contributions from intra-band transitions, exciton-induced bandshifts (EIS)[34] and exciton-induced dephasing (EID)[35–37]. Partial overlap of the ESA bands with the GSB makes it difficult to disentangle the corresponding contributions (Supplementary Figs. 10, 11). At the first 1 ps, the GSB bands show a slight red-shift (~3 nm), reflecting hot carrier cooling (Supplementary Fig. 12a, b). With increasing excitation density, the main band-edge bleach shows an obvious blue shift because of the Burstein-Moss effect (Supplementary Figs. 12c, 13a)[38,39]. The previous studies demonstrate that the bandgap renormalization due to the photogenerated charge carriers, a so-called excitation-induced shift (Supplementary Fig. 13b), can also affect the bandgap of the material, typically resulting in a red-shift of GSB peaks[40,41]. These two effects partially compensate each other, resulting in the observed slight GSB red-shift. We here exclude the possibility of a direct-to-indirect bandgap transition since the 2D system shows a direct bandgap and the GSB red-shift appears promptly (within 100 fs), scaling with excitation density (more details see Supplementary Note 3, Fig. S12). In Fig. 2d, the TA spectrum of the 1D system shows a broad ESA signal across the probe wavelength region. Such a spectrum is consistent with the STE state seen in earlier studies[42,43].

By using the singular value decomposition (SVD) global analysis (GLA) method, we now quantify the excited state dynamics in these materials. In the 2D system, we obtain four components: $\tau_1 \sim 210$ fs, $\tau_2 \sim 56$ ps, $\tau_3 \sim 620$ ps, and $\tau_4 > 5$ ns (see Supplementary Fig. 14). The ESA signal at > 625 nm rapidly decays with a few hundred fs lifetime, suggesting that the ultrafast component $\tau_1$ corresponds to the hot carrier cooling. The intensity-dependent TA measurements reveal that the lifetime of the component $\tau_1$ becomes longer at high excitation intensities due to the hot phonon bottleneck (Fig. 2, Supplementary Figs. 14–16 and Table S2)[3]. Conversely, the time constants $\tau_2$ and $\tau_3$ decrease dramatically. We explain this behavior in terms of general second and third-order decay. Indeed, at the lowest excitation intensity, the band-edge GSB kinetics show a linearity of $\triangle A^{-1}$ as a function of delay time at first 600 ps resulting from the second order process, while at higher intensities they show the linearity of $\triangle A^{-2}$ vs delay time (indicative for the third order process), see Supplementary Fig. 17[38,39]. Given this behavior, the longer decay components should not be directly interpreted as simple well-defined rate processes but rather reflect a combination of the second-order Saha-Langmuir type carrier recombination (evidenced by $\triangle A^{-1}$ dependence), the third-order Auger recombination at higher excitation intensities (evidenced by $\triangle A^{-2}$ dependence), and carrier-trapping by the surface states (evidenced by the low-energy emission tail in the steady-state emission spectrum). In the 1D system, the GLA results also give four components, but with notably different time constants (Supplementary Fig. 18). The precise value of the fastest component of 480 fs has

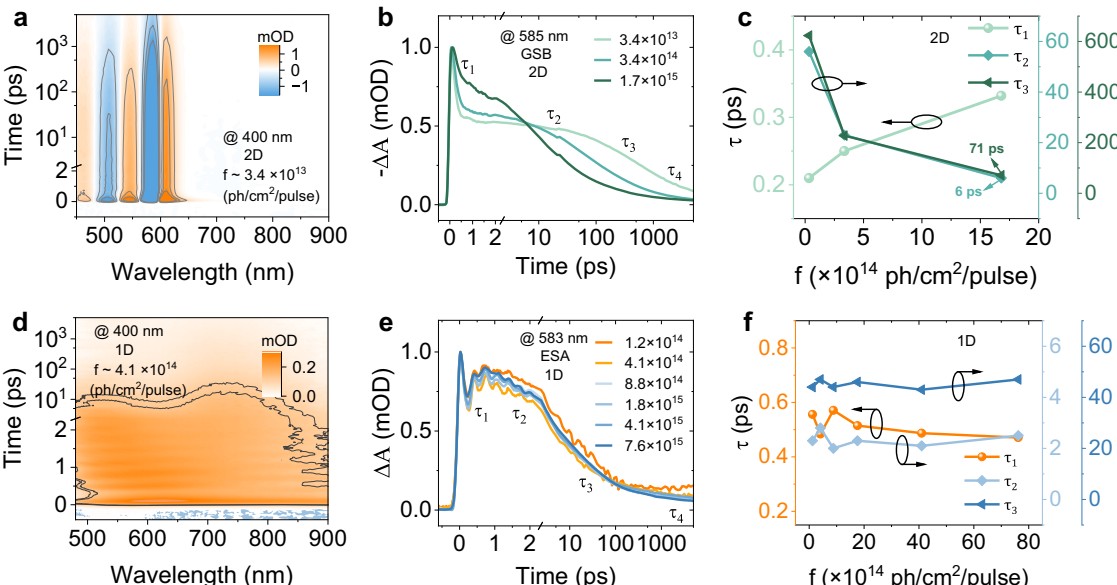

**Fig. 2 | Exciton dynamics at different excitation fluences.** Pseudocolor representation of fs-TA spectra, temporal kinetics, and time constants extracted from SVD analysis of **a**–**c** 2D ODASnI$_4$ and **d**–**f** 1D ODASn$_2$I$_6$ thin films. The excitation fluences are shown in the figures with the unit of photon/cm$^2$/pulse.

considerable uncertainty since the first half-phase of the pronounced oscillatory signal on the same timescale disturbs it. Still, it is clearly present. We follow the model of two STE states in our previous work with refinements based on the new data and closer analyses[26]. For the fastest component, we need to consider the nature of the initial state directly after light absorption. The tightly packed 1D structures likely support coherent delocalization over at least a few wires, then, because of the strong EPC, we expect that the initially delocalized state rapidly loses coherence and localizes analogously to the formation of excitonic polarons in molecular complexes[44]. We therefore assign the fastest component to the excited state localization, together with the cooling and the initial phase of the STE formation. The assignment is further supported by the similarity in spectral shape between the two fastest components and the observed spectral shift to higher energies. This is expected from an ESA signal reflecting the initial state relaxing to lower energies, thereby increasing the ESA transition energy. The second component of 2.8 ps converts the initial broad single-band spectrum to a rather different form with two bands, indicating the formation of a new state – STE$_2$. The two longest components have similar spectra, suggesting no further change in state character. The longest component of ≫ 5 ns has half the amplitude of the 47 ps component. This, together with the PL quantum yield of 37 % makes us suggest that the 47 ps component corresponds to a nonradiative decay channel due to some structural peculiarities that affect about 60% of the excitations. The longest component (≫5 ns) represents the remaining excitations that decay radiatively, consistent with the observed long PL lifetime (Supplementary Fig. 5b). The ns-TA spectrum presents the broad ESA signals across the whole wavelength detection window and the decay probed at 583 nm lasts over 2 μs, which is well in line with the TRPL measurements (Supplementary Fig. 19). Therefore, we assign this component to the radiative recombination of the long-lived STE state. The kinetics of the 1D system do not show almost any intensity dependence – this result is also very different compared to the 2D system (Supplementary Figs. 20–24). It means that the localized STE state is so well shielded by the deformation of the local environment that they do not interact with each other even at the excitation density over $2.9 \times 10^{20}$ excitation/cm$^3$/pulse. Furthermore, the PL intensity increases linearly with the excitation fluences (Supplementary Fig. 25), demonstrating that no additional nonradiative recombination channel appears in the 1D system at

higher excitation intensities. It also means that the PL in this system is mostly the first-order process rather than the second-order electron-hole pair recombination as in the bulk perovskite microcrystals[45].

## Vibrational dynamics

The ultrashort laser pulse can excite coherent vibrational wavepackets of the modes coupled to the electronic transition. Their generation mechanism can be described by a damped harmonic oscillator driven by the external force (for details, see Supplementary Note 4). The wavepackets oscillate on the potential energy surface with certain mode frequencies, and the amplitude of these oscillations strongly depends on the exciton-phonon coupling strength in the materials. As shown in Fig. 2, a clear oscillatory signal appears in the 1D system at room temperature. Interestingly, no oscillations can be observed in the 2D system. Considering that the EPC and phonon dephasing are subject to temperature, we also conduct fs-TA measurements at 77 K and 200 K. The time-wavelength spectrograms of differential absorption are shown in Fig. 3a, b, and Supplementary Figs. 27, 28. We find that at 77 K, oscillations appear even in the 2D system. Since in this case we excite high above the lowest band, most likely creating free charges, the oscillations may be partially driven by the charge carrier coupling to the phonons. However, the carrier cooling happens with a time constant of 210 fs, which is much faster than the phonon oscillation period. Therefore, we conclude that the exciton-phonon coupling dominates the phonon coherence in the 2D system. After subtracting the population dynamics from TA signals, we derive beating maps of oscillatory components (Fig. 3c, d and Supplementary Fig. 29). The vibrational modes are analyzed from the oscillatory signals in terms of a sum of damped cosine functions

$$\Delta A_{\text{osc}} = \sum_{i=0}^{n} A_i \exp\left(-\frac{t}{\tau_i}\right) \cos(2\pi v_i t + \varphi_i) \quad (1)$$

where $A$, $\tau$, $v$ and $\varphi$ are the amplitude, dephasing time, mode frequency, and phase of the $i^{\text{th}}$ oscillating component, respectively. The fitting results, presented in Supplementary Figs. 27, 28, 30, Table S3, 4, reveal that the vibrational modes that appear in the TA dynamics are very different in the two systems. The dephasing time of the dominant phonon mode in the 2D system is over 1.4 ps, which is approximately 2.5 times longer than in the 1D system (-0.53 ps). By Fourier

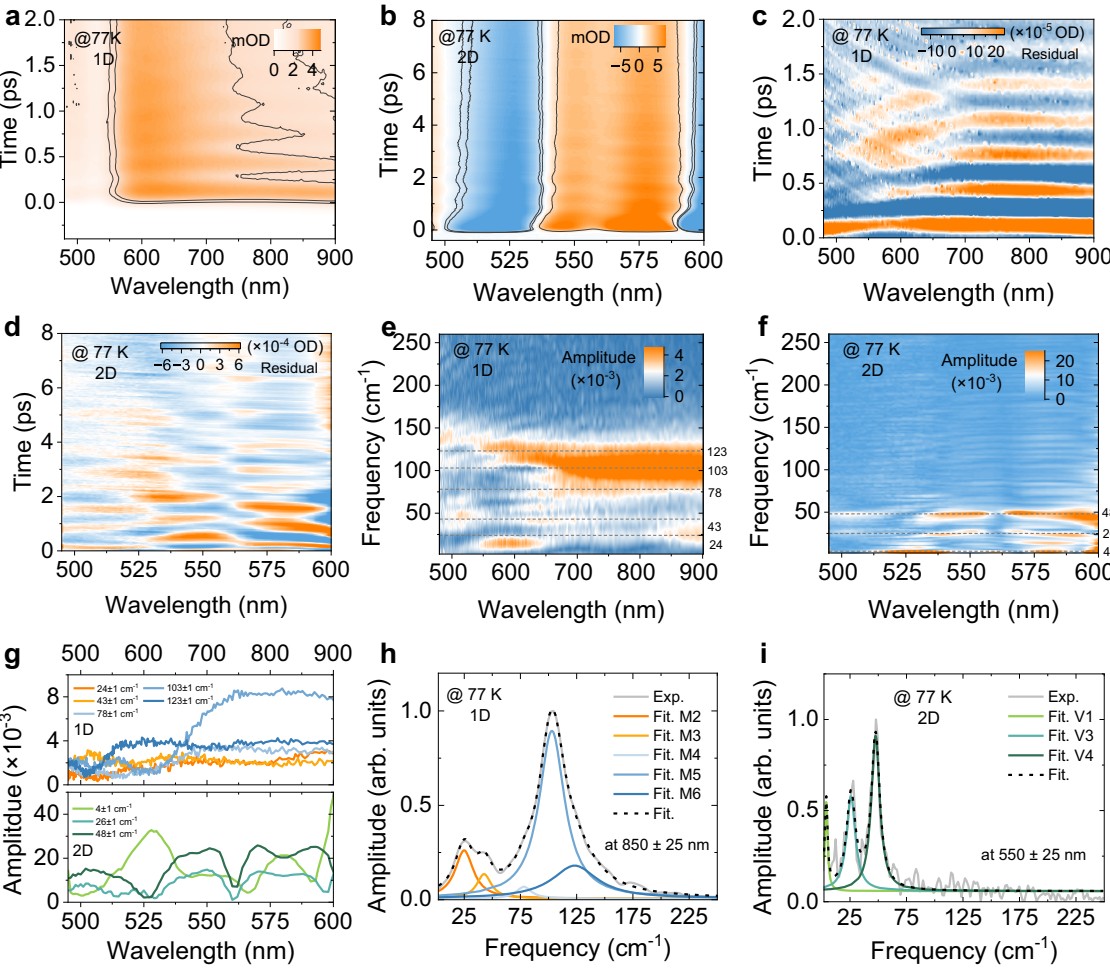

**Fig. 3 | Vibrational mode analysis of oscillatory wavepacket at 77 K.** Pseudocolor representation fs-TA spectra of **a** 1D and **d** 2D systems. Beating maps of oscillatory components after subtracting the exponential decay: **b** 1D system; **e** 2D system. Probe wavelength resolved vibrational frequencies of **c** 1D and **f** 2D systems directly obtained by Fourier transforming the differential TA spectrum. **g** Slices of Fourier transforming the oscillatory signals, we uncover the frequencies of vibrational modes and the amplitude of oscillations across the probe wavelengths, presented in Fig. 3e–i, Supplementary Figs. 31,32, and Table S5. The fitted vibrational spectrum allows us to assign five modes in the 1D system: M2 (25 cm$^{-1}$), M3 (43 cm$^{-1}$), M4 (78 cm$^{-1}$), M5 (103 cm$^{-1}$), and M6 (123 cm$^{-1}$). At higher temperature, a low-frequency mode M1 (15 cm$^{-1}$) appears, and a slight upwards shift of the mode frequencies takes place, accompanied by the broadening of the bands due to the enhanced phonon-phonon scattering. Especially, the M5 frequency changes to 106 cm$^{-1}$ at room temperature. In comparison, the 2D system shows only three vibrational modes at 77 K, all below 50 cm$^{-1}$, with the most intense peak at 47 cm$^{-1}$. The V1 (3 cm$^{-1}$) represents the very low-frequency phonon wing and is probably partially overlapping with the DC component. No such low-frequency component is observed in the 1D system. All modes in the 2D system are strongly damped and not visible at room-temperature dynamics. We propose that the significant differences in vibrational modes of 1D and 2D systems can be due to the variations in ligand concentration, resulting in a distinct degree of lattice distortion and phonon anharmonicity[21].

We mention that the EPC strength estimation of 2D ODASnI$_4$ from the temperature-dependent PL spectra is not reliable because of the emission of a surface trap state. To facilitate the comparison of EPC strength in these systems, we estimate the Huang-Rhys factor $S$ for the

transformed beating maps at dominant mode frequencies. Vibrational spectra with the Lorentzian fitting results: **h** 1D system (probed at 850 ± 25 nm); **i** 2D system (probed at 550 ± 25 nm). V and M are used to distinguish the modes in these two systems.

2D system by adopting a displaced harmonic oscillator model that has been used for quantum dots and perovskites to obtain the EPC strength from the coherent wavepacket TA signal[12,21,46–48]. In this analysis, the EPC strength is proportional to the amplitude of the oscillatory signal ($\Delta A_{osc}$), which is related to the lattice reorganization energy ($\lambda$) through $\Delta A_{osc} = \lambda \cdot \frac{dOD}{dE}$ (for details see Supplementary Note 4). We focus on the vibrational modes at the band-edge GSB signal (595 nm). Using $\frac{dOD}{dE}$ from the normalized absorption spectrum, we obtain the $S$ factor for all the modes, and the results are summarized in Supplementary Table 6. The calculated $S$ factors of the V3 and V4 modes are 0.8 and 1.3, respectively. This method cannot be used for the 1D system since we do not observe the GSB bands in the detection window. Given the previous results, we note that the exciton-phonon interaction of the 2D system is drastically smaller than that of the 1D system – the lattice reorganization energy is >30 times lower. Such a significant difference can be attributed to enhanced disorder-induced Anderson localization in the 1D system[49–52]. The effect is further enhanced by the inherently larger disorder and softness of the 1D system, the former enhancing the localization, the latter making the localization response, as the lattice deformation, stronger.

To elucidate the origin of the strong EPC in the 1D system and assign the experimentally observed coherent vibrational modes, we perform the first-principles calculations. Density functional theory (DFT) calculations are carried out in Vienna ab initio Simulation

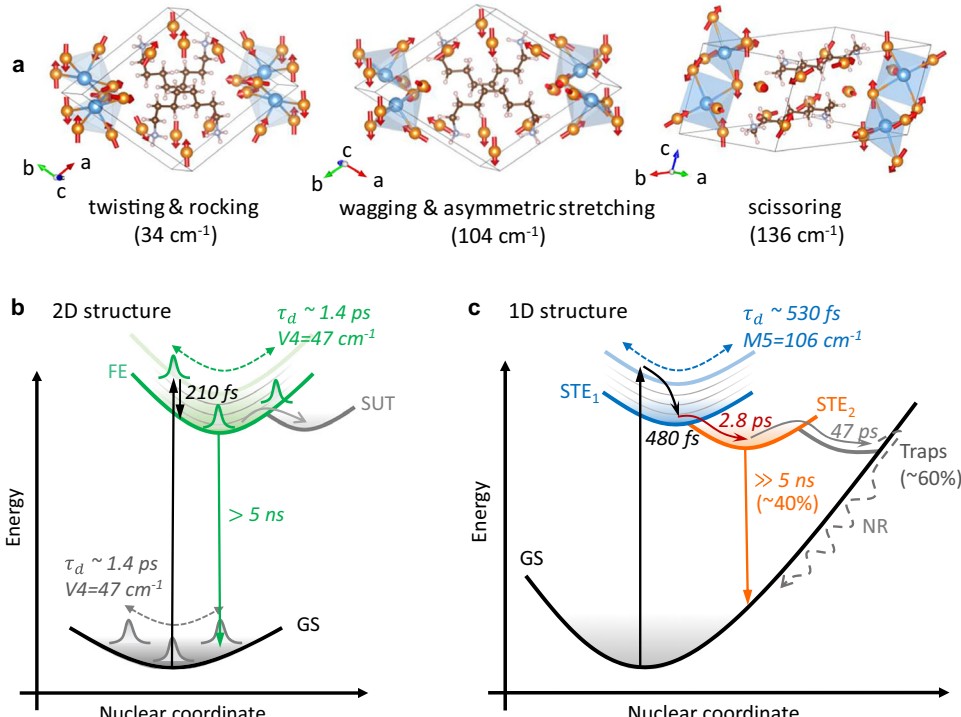

**Fig. 4 | Summary of vibrational dynamics and electronic processes. a** Selection of calculated vibrational modes and the corresponding atomic motions in ODASn$_2$I$_6$. Schematic illustrations of vibrational and electronic dynamics: **b** 2D ODASnI$_4$ and **c** 1D ODASn$_2$I$_6$ thin films. FE: free exciton; STE: self-trapped exciton; GS: ground state; SUT: surface trap states. The ultrashort pulse excitation initiates coherent wavepackets on the electronic states via ISRS (oscillation on the ground state) or DECP (oscillation on the excited state). The damping time of each dominant mode is denoted by $\tau_d$, with the mode frequencies labeled M for the 1D system and V for the 2D system. The dominant mode M5 of the 1D system is assigned to the 104 cm$^{-1}$ mode in the calculations. Dashed double arrows depict oscillatory wavepacket propagation. In the 2D structure, the high-energy excitation populates the hot FE state that undergoes carrier cooling (~210 fs). Some carriers are trapped to form the surface trap state and the second-order recombination, respectively. In the 1D structure, the initial rapid dynamics consist of carrier cooling and self-trapped exciton formation combined with wavepacket dynamics (~480 fs). Electronic relaxation happens between the two STE states. After that, ~60% of excitations decay nonradiatively, while the remaining ~40% contribute to PL emission.

Package (VASP) with finite differences approach and density functional perturbation theory (DFPT)[53]. In addition, the temperature-dependent velocity autocorrelation functions (VACF) are obtained from the Ab initio molecular dynamics (AIMD) simulations. The resulting vibrational density of states (VDOS) spectrum exhibits prominent peaks below 250 cm$^{-1}$, with the strongest peak in the range of 80–110 cm$^{-1}$ (Supplementary Fig. 33). Consistently, the FFT analysis of the electronic bandgap autocorrelation function in Fig. 1 also reveals a dominant vibrational mode at 104 cm$^{-1}$, well matching the M5 mode obtained from the TA dynamics. The electronic transition mainly involves the tin and iodide atoms, suggesting these modes are closely related to the atomic motions in tin iodide units (Supplementary Figs. 34, 35). The temperature-dependent analysis of VACF further shows a shorter dephasing time of these modes at elevated temperatures, smoothening away the distinct vibrational lines present at 77 K. Such vibrational energy redistribution and dephasing reflect the anharmonic nature of these vibrations. The vibrational mode assignments in the 1D system are based on the finite differences calculations of the inorganic framework by excluding the organic component, since the inclusion of a large amount of ODA ligands in one unit cell introduces a significant challenge in convergence and increases mode complexity. We thus discuss the lattice vibrations considering the closest correspondence with experimental results. The vibrational modes at 23, 51, and 136 cm$^{-1}$ are attributed to pure rocking, twisting, and scissoring of I–Sn–I, respectively. The other modes represent mixed I–Sn–I motions: twisting with rocking (34 cm$^{-1}$), twisting with asymmetric stretching (77 cm$^{-1}$), I-Sn-I wagging with asymmetric stretching (104 cm$^{-1}$), see Fig. 4a, Supplementary Fig. 36, and Table 7.

## Discussion

The coherent phonons can be generated either displacively in the excited state or impulsively in the ground electronic state. In both cases, to be visible in TA as oscillations, the potential energy surfaces (PES) of the corresponding modes need to be displaced. Initiation of the excited state wavepacket is thereby sometimes called displacive excitation of coherent phonons (DECP). Since the phonons start at maximum amplitude, they show cosine-like oscillations[35,43,54]. The ground state wavepacket is explained by the impulsive stimulated Raman scattering (ISRS)[22,40,41,55]. In this case, the wavepacket starts with negligible initial amplitude and thereby shows sine-like oscillations. The initial phase of coherent oscillations indicates which excitation pathway dominates in the excitation of the wavepacket[48]. As shown in Supplementary Fig. 37a, the oscillatory signal at 595 nm in the 2D system presents an initial phase of π, which falls in the Franck-Condon region evolving on the excited state PES. Along with the propagation of the excited state wavepacket, the phonon oscillation at 550 nm shifted by π out of phase relative to that at 595 nm, indicating that the wavepacket has passed through the equilibrium position on the PES of the FE state and is now oscillating on the opposite side. Additionally, the V4 mode at 595 nm (GSB) and 550 nm (ESA) presents the phase of 1.8π and 0.64π, which are between the sine- and cosine-like oscillation most likely originating from their combination (Supplementary Table 3). In our experiments, the 2D sample was excited above the bandgap, then the vibrational wavepackets could form on both the ground and excited states. It means that the oscillations in the 2D system generally originate from both ISRS and DECP. The schematic vibrational dynamics are shown in Fig. 4b. The fitted dominant mode frequencies

(V4) at 595 nm and 550 nm have a difference of 2 cm$^{-1}$ falling in the range of frequency uncertainty (3 cm$^{-1}$); therefore, we present that the vibrational wavepackets propagating along the PES of the ground state and FE state are the same. In TA measurements of the 1D system, we only observe the ESA signals. The oscillatory signals at 500 and 851 nm are almost perfectly in phase (Supplementary Fig. 37b). Taken together, this suggests that the vibrational wavepackets reside on the excited state induced via DECP. In the oscillatory signal analyses, we notice that the M5 mode (106 cm$^{-1}$) is dominant with the highest amplitude. The Fourier transform of the bandgap fluctuation autocorrelation function is dominated by a mode of very similar frequency (104 cm$^{-1}$), which gives the largest contribution to the reorganization energy. Therefore, we conclude that the corresponding vibrational wavepackets propagate along the STE formation potential energy surface, and the M5 mode is related to the lattice distortion, driving the system to be strongly self-trapping, see Fig. 4c.

The role of organic cations in modulating lattice distortions and exciton–phonon coupling is well established in lead-based 2D perovskites[9,11,12,23,25,56–63]. Extending this framework, our study shows that even without changing the chemical properties of the organic cations, tuning the dimensionality from 2D to 1D via ligand concentration control leads to pronounced differences in exciton localization and phonon coupling. While the 2D system exhibits robust excitonic features and FE emission consistent with quantum and dielectric confinement[64], the 1D system shows much stronger exciton localization resulting in pronounced STE emission. The excitation-intensity dependent TA measurements reveal that the STE population dynamics remain largely unaffected by changes in excitation conditions; however, the FE dynamics are significantly influenced by hot carrier cooling and Auger recombination, particularly at higher excitation intensities. The temperature-dependent TA measurements further demonstrate that the coherent vibrational wavepackets on the STE state in the 1D system are dominated by a phonon mode at 106 cm$^{-1}$ involved in the wagging and asymmetric stretching vibration in [SnI$_5$]$^-$, while the coherent vibrational wavepackets on the FE state and the ground state in the 2D system have significantly lower frequencies below 50 cm$^{-1}$.

In summary, our combined experimental and computational analysis uncovers the critical role of dimensionality-dependent exciton–phonon interactions in governing both the lattice dynamics and electronic transitions, including exciton self-trapping, in organic-inorganic tin halide systems. This modulation, achieved simply through concentration-controlled tuning of the organic components, provides fundamental insight into the relationship between the structure and functionality in this emerging class of materials. Moreover, it establishes a framework for ligand-mediated control of exciton-phonon coupling, opening new avenues for tailoring the optoelectronic properties of lead-free hybrid metal halides for sustainable photonic and energy applications.

## Methods

### Thin film preparation and characterization
The Octylenediammonium diiodide (ODA-I$_2$) is synthesized by following steps: 1,8-Diaminooctane (1500 mg, 10.4 mmol) was dissolved in ethanol (2 mL), followed by slow addition of HI (1.5 mL, 20 mmol). The solvent is evaporated under reduced pressure to give a beige solid, which is vacuum-filtered, washed with CH$_2$Cl$_2$, and dried at 120 °C under vacuum for 3 h to afford DAO–I$_2$ as a white powder[27]. Then ODA-I$_2$ and SnI$_2$ are dissolved in anhydrous DMF at volume ratios of 2:1 or 4:1 to prepare precursor solutions with a concentration of 1 mol·mL$^{-1}$. The nm-scale thin films are spin-coated from the prepared precursor on a clean substrate and annealed at 75 °C for 8 minutes. More details are illustrated in the Supplementary. The film's general morphology is characterized using a Philips XL30FEG scanning electron microscope (SEM) operated at 3 kV. Powder X-ray diffraction (XRD) measurements are performed on a Pananalytical X'Pert Pro diffractometer equipped with a Cu Kα X-ray source ($\lambda = 1.5406$ Å). Steady-state absorption spectra are recorded using a UV-vis spectrophotometer (PerkinElmer Lambda 1050). Photoluminescence (PL) spectra and time-correlated photokinetics are obtained with a Horiba Spex 1681 spectrometer, using 375 nm excitation. For temperature-dependent PL, measurements are taken during heating from 77 to 298 K. PL decay dynamics of ODASn$_2$I$_6$ are analyzed with a custom-built time-correlated single-photon counting (TCSPC) setup, featuring Silicon single-photon avalanche diodes (Si SPADs, IDQ100, quantum sensing), a time-tagging device (quTAG), and a 375 nm pulsed laser (EPLED Series, 100 kHz to 2.5 MHz). The system's time resolution is approximately 40 ps. PL decay dynamics of ODASnI$_4$ are analyzed home-built TCSPC setup with a 405 nm picosecond pulsed laser (laser diodes, LDH-series, Pico-Quant) for excitation. The adjustable repetition rate is 2.5 MHz with a pulse duration of 20 ps. The single photon avalanche diode (SPAD, PicoQuant) is used as a detector with a timing resolution as short as 50 ps.

### Femtosecond transient absorption measurements
Femtosecond to nanosecond transient absorption (fs-ns TA) measurements are performed on a custom-built femtosecond pump–probe spectroscopy. The laser source is a Solstice regenerative amplifier (Spectra Physics; 8 W, 796 nm, 60 fs, 4 kHz) seeded by a Mai Tai SP femtosecond oscillator (Spectra Physics). The seedling laser is split into a pump beam (95% of full power) and a probe beam (5% of full power). For the probe, the 1350 nm pulses generated by a collinear optical parametric amplifier (TOPAS-C, Light Conversion) are focused on a thin CaF$_2$ plate to generate supercontinuum white light, covering the visible spectrum from 480 to 900 nm. The probe beam is then split into two parts: one passes through the sample, and the other is sent directly to the detector for reference. The pump pulse (70 mW, 400 nm, 100 fs) is produced by another collinear optical parametric amplifier (TOPAS-C, Light Conversion), followed by frequency doubling of the 800 nm pulses. The polarization angle between the pump and probe beams is set to the magic angle (54.7°). The excitation intensities are changed in sequence by using the neutral density filter. To prevent photodamage to the samples, the sample position is changed after each scan. For temperature-dependent fs-TA measurements, samples are mounted in an Oxford cryostat under vacuum to minimize moisture exposure. The excitation fluences for ODASn$_2$I$_6$ and ODASnI$_4$ are fixed at ~1.2 × 10$^{16}$ and ~2.0 × 10$^{14}$ excitation/cm$^2$/ pulse. Global analysis (GLA) is carried out in the Glotaran software package (http://glotaran.org), employing singular value decomposition (SVD) for global fitting incorporating multiple components[65].

### Computational details
All calculations are carried out with Vienna Ab initio Simulation Package (VASP) using the Projector Augmented Wave (PAW) method to solve the Kohn-Sham equations[66,67]. The exchange–correlation potential is treated using the Perdew–Burke–Ernzerhof (PBE) generalized-gradient approximation (GGA) in a non-spin-polarized approach[67]. A plane-wave kinetic-energy cutoff of 520 eV is used throughout the electronic-structure calculations. The Brillouin zone sampling for the electronic structure calculations is performed using a reciprocal grid of 2 × 2 × 3. For structural optimization, a 1 × 1 × 2 reciprocal grid is employed. Geometries are considered converged when the residual forces are below 0.001 eV·Å$^{-1}$, and the total-energy change is less than 10$^{-5}$ eV. To obtain a more reliable bandgap by reducing the known deficiencies of semilocal functionals, the hybrid Heyd–Scuseria–Ernzerhof (HSE06) functional is used[68]. Lattice vibrational properties are evaluated using a finite-displacement approach. Mode analysis is performed at the Γ point, using a tighter electronic convergence of 10$^{-8}$ eV and a reduced plane-wave cutoff of 400 eV for this step.

Ab initio molecular dynamics (AIMD) simulations using the PBE functional are performed to further investigate lattice dynamics. The time-correlated energy bandgap simulation is first computed from a 2 ps trajectory propagated with a 20 fs time step. To investigate possible slower vibrational dynamics, the same bandgap-correlation analysis is repeated along extended trajectories up to 20 ps, using a 1 ps sampling step and including Grimme's D3 dispersion corrections together with spin–orbit coupling (SOC). The temperature-dependent velocity autocorrelation function (VACF) simulation runs over 20 ps with a time step of 1 fs (NVT ensemble, Grimme's D3 dispersion corrections and SOC). The vibrational density of states (VDOS) is derived from the atomic velocity trajectories in the NVT ensemble. The VACF and VDOS data processing is done by VASPKIT[69,70]. For each atom $i$, the velocity autocorrelation function is evaluated as[70–72]

$$c_i(\tau) = \langle \mathbf{v}_i(\tau) \cdot \mathbf{v}_i(0) \rangle \qquad (2)$$

Where $\mathbf{v}_i(\tau)$ denotes the velocity of the $i^{\text{th}}$ atom at time $\tau$. The angular brackets denote an average over the entire simulation trajectory. For convenience of comparison, the VACFs are normalized such that $c_i(0) = 1$ in the output. To obtain the overall VACF of the system, the contributions from each atom are combined through a mass-weighted summation:

$$C(\tau) = \sum_{i=1}^{N} m_i c_i(\tau) \qquad (3)$$

Where $N$ is the number of atoms and $m_i$ is its atomic mass. Within the harmonic approximation, the vibrational density of states is then obtained from the Fourier transform of $C(\tau)$:

$$f(\omega) = \frac{1}{k_B T} \int_{-\infty}^{\infty} C(\tau) \exp(-i\omega\tau)\, d\tau \qquad (4)$$

where $\omega$ is the vibrational angular frequency, $k_B$ is the Boltzmann constant, and $T$ is the absolute temperature. At $\tau = 0$, $C(0)$ is proportional to the total kinetic energy of the system, $C(0) \propto k_B T$, according to the equipartition theorem. The prefactor $1/(k_B T)$ is introduced for normalization purposes and serves to remove the explicit temperature dependence of the mass-weighted VACF. In the present work, we omit this constant prefactor and focus on the spectral shape; therefore, the VDOS is expressed as

$$S(\omega) = \int_{-\infty}^{\infty} C(\tau) \exp(-i\omega\tau)\, d\tau \qquad (5)$$

## Data availability
The data that support the findings of this study are provided in the main text and the Supplementary Information. Data supporting the main-text figures are available via Figshare at https://doi.org/10.6084/m9.figshare.30604079. More data are available from the corresponding author upon request.

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

## Acknowledgements

This work was supported by the Swedish Energy Agency (Grant 50667-1, 50709-1), the Swedish Research Council VR (2021-05207, 2023-05244), Olle Engkvist Foundation (Grant 235-0422), Knut and Alice Wallenberg Foundation (Dnr KAW 2019.0082), and the China Scholarship Council (No. 202006150002). The computations were enabled by resources provided by the National Academic Infrastructure for SuperComputing in Sweden (NAISS) via the project 2024/5-372. Collaboration with NanoLund is acknowledged. Y.W. acknowledges the Postgraduate Research & Practice Innovation Program of Jiangsu Province (Grant no. KYCX23_0368). J. C. acknowledges the Novo Nordisk Foundation (Grant no. NNF22OC0073582).

## Author contributions

T.P. and Y.H. conceived and initiated the project. Y.H. carried out steady-state/ultrafast experiments and validated and analysed the experimental data. F.G. and X.C. prepared the samples, performed XRD and TRPL measurements. M.Z. performed SEM/EDS measurements. R.A. and T.E. carried out the band structure and phonon calculation. Y.W. conducted AIMD calculations. S.R. contributed to the discussion on the coherent phonon generation mechanism. J.C. contributed to the discussion on oscillatory signals. T.P. and Y.H. extensively discussed the data and formulated the conclusions. Y.H. prepared the first draft of the manuscript. All authors contributed to the revision and editing of the manuscript.

## Funding

## Competing interests

The authors declare no competing interests.
