## [Transparent Peer Review file · Nature Communications]

Dimensionality-dependent electronic and vibrational dynamics in low-dimensional organic-inorganic tin halides

Corresponding Author: Professor Tõnu Pullerits

Version 0:

Reviewer comments:

Reviewer #1

(Remarks to the Author)

The study by He et al. reports very interesting results on dimensionality-dependence of excitonic effects in 1D and 2D metal halide perovskites. The findings are very interesting and the manuscript well written and presented. It seems to be a good fit for the journal, the following major comments require further clarifications:

1. from the manuscript it was not entirely apparent or discussed in depth why dimensionality would affect exciton localization. For a single electron, the situation is quite clear in the framework of Anderson localization. For excitons this might be similar. Is this the reasoning? This should be discussed in more detail in the manuscript and the novelty aspect of the study should be made more clear in this context.

2. the theoretical simulations using AIMD method were not appropriately performed. The authors stated that a 5ps run was used, yet their VDOS (and the experiments) clearly show that dynamics slower than 1ps are present. It is recommended that the authors demonstrate convergence of the VDOS using at least 20 ps of runtime, which is what this reviewer would expect is required to obtain statistically converged data. Moreover, what functional was used during the MD run, and did the authors test the effect of dispersion corrections? These corrections are of key importance in these systems and may significantly impact the reported data. Furthermore, when band gaps were calculated, did the authors consider spin-orbit effects and, if not, how is this justified in these materials? Finally, in the phonon calculations did the authors observe any imaginary modes? These materials are prone to show imaginary modes because there are phase transitions: which structure has been used in these calculations, is it the ground-state structure and were there any imaginary modes present?

Reviewer #2

(Remarks to the Author)

The manuscript by Y. He et al reports a study of 1D and 2D tin-halide perovskites addressing in particular exciton-phonon couplings by photoluminescence and transient absorption spectroscopy. Emission from 2D samples is found to be dominated by free excitons, whereas in the 1D case self-trapped excitons dominate. Exciton-phonon coupling is observed to be ~2.5 times larger in 1D compared to the 2D structure. Phonon modes are assigned based on DFT calculations (VASP). The manuscript is generally well written.

Both optical and electronic properties of halide perovskites are generally strongly influenced by the interaction with the polar lattice. Additionally, the presence of organic spacers in 2D systems gives rise to unique and complex optical properties, so far not well understood yet.

Tin halides represent a promising lead-free material platform and are thus highly interesting for applications although they are known to be more prone to stability issues. Importantly, low dimensional tin halide perovskites are significantly less thoroughly investigated than lead-based counterparts. As such this study reports a highly relevant and timely topic and can contribute an important step in filling the gap of knowledge on optical excitations and lattice behavior of low-dimensional tin-halide perovskites.

Despite the relevance of the work, I have a few concerns and questions (reported below) that need to be addressed before a decision on publication can be made.

1) One of the main findings of the manuscript is that the 1D system shows much larger exciton-phonon coupling than the 2D system for the dominantly coupled mode. While this result is plausible due to stronger exciton localization in 1D, the manuscript currently lacks a detailed physical explanation for this finding.

2) The role of the organic spacer in modulating inorganic lattice distortion, and consequently the exciton-phonon interaction having considerable impact on optical properties, has been previously proposed for 2D perovskites (e.g. <https://doi.org/10.1016/j.trechm.2019.04.003>). The present manuscript supports that these mechanisms may be analogous in tin-based systems.

3) When exciting at 400 nm, likely free charge carriers, rather than bound excitons, are predominantly initially generated. How do the authors ensure that the vibrational modulations observed in the nonlinear TA signal originate from exciton-phonon rather than charge-phonon coupling?

4) In the TA measurements of the 2D sample, two GSB and three ESA bands are observed. The observed red-shifts of the GSB bands are explained in terms of initial carrier cooling and excitation dependent effects such as EIS. The origin of the ESA bands in contrast is not clearly discussed in the manuscript.

Moreover, lead-halide perovskite have been reported to be sensitive to both excitation induced shift and excitation induced dephasing (<https://doi.org/10.1021/acs.jpcc.5b00148> ; <https://doi.org/10.1146/annurev-physchem-102822-100922>; <https://doi.org/10.1021/acs.jpcc.2c00658>). EID arises from pump-induced broadening of the exciton resonance, that can be due to exciton-charge carrier or exciton-exciton interactions at increasing high fluence. These can result in typical lineshape distortions and appearance of side bands in nonlinear experiments, such as TA, and have been widely investigated both experimentally and theoretically in conventional semiconductors. Can the authors exclude here that the lineshapes are not affected by EID?

Since experiments at different excitation density are performed, an excitation-density-dependent lineshape analysis could help to check this point. Additionally, it should be ensured that the experiments are performed at carrier densities below the Mott transition (<https://doi.org/10.1038/s41467-020-14683-5> ; <https://doi.org/10.1021/acs.jpcclett.9b01936>).

5) For the 2D system presented in this manuscript, two distinct excitonic peaks are observed in the absorption spectrum at 502 nm and 588 nm. The physical origin of these transitions should be addressed. Are these related to different exciton species, or polaronic states?

6) The manuscript does not mention the excitonic structure or binding energy for the 2D and 1D systems investigated in this work. 2D perovskites exhibit pronounced exciton fine-structure splitting of band-edge excitons with bright-bright splitting of a few meV, dark-bright splittings around 10 meV, and in some Br-based materials even tens of meV. Fine structure appears important in general for light-matter interaction of these quasiparticles (DOI: 10.1126/sciadv.abk0904; <https://doi.org/10.1063/5.0048490>; <https://doi.org/10.1021/jacs.3c11957>;). Estimating exciton binding energies and discuss the excitonic structure of these materials would enrich the analysis.

7) As a general remark, I strongly recommend to put the discussion in the context of the large body of literature that is available for low dimensional lead-halide counterparts highlighting potential similarities and differences with the present findings on tin-based 2D and 1D structures.

Minor comments:

Fluences are reported without units in Fig. 2 and related Supplementary Figures.

With respect to vibrational dynamics, I recommend showing residual maps in the manuscript, e.g. as additional panels in Fig 3, not only in the SI, since these maps are essential for the results.

In Fig. 3 panel c and d the labels for y-axes (Frequency (cm⁻¹) ?) are missing.

Reviewer #3

(Remarks to the Author)

The work by He et al. reports on the role of the sample dimensionality on the exciton-phonon interactions in thin-layered Sn-halide semiconductors. Previous work by (part of) the authors reported on a self-trapped exciton for OD ASn₂I₆, which is absent for the 2D counterpart OD ASnI₄. With the use of ultrafast transient absorption spectroscopy, the dimensionality-dependent charge carrier dynamics are studied as a function of temperature, to elucidate the lattice dynamics in low-dimensional tin-based semiconductors. By taking the fast Fourier transform of the lower-temperature fs-TA data, I-Sn-I vibrational modes are identified which are used to rationalize charge carrier dynamics observed for both 1D and 2D systems.

Overall, I have some issues with the novelty of the work. A large part of the steady-state and transient spectroscopy experiments (Figs. 1a,d,f and Figs. 2d-f, Fig. 4c) appear to be reproduced from earlier work by the group [Adv. Optical Mater. 2025, 13, 2402061]. Additionally, the authors conclude that the free exciton dynamics in 2D thin films are significantly influenced by hot carrier cooling and higher-order recombination, which is well established for inorganic-organic

semiconductors. This is reducing the novelty of this work to the detailed analysis on the vibrational modes extracted from the low-temperature TA experiments. As such, I unfortunately cannot recommend this work for a high-impact broad audience outlet like Nature Communications and would suggest publication of this additional (but nice) data set in a more specialized outlet.

Specific comments:

1. Can the authors comment on the self-absorption (if any) for OD ASnI₄? The asymmetric shape of the PL spectrum suggests there is a considerable self-absorption. As panel a suggests a considerable overlap of the PL and absorption spectra, how does this affect the determined FWHM used to determine the Huang–Rhys (S) factor?
2. On page 3, second paragraph, it is mentioned the fs-TA experiments were performed under resonant excitation conditions. In the manuscript it is mentioned that the absorption peaks are at 502 nm and 528 nm. However, a 400 nm pump pulse is used. Can the author clarify why they excite so off-resonance at 400 nm?
3. With the use of electronic structure calculations, the time-dependent bandgap energy is determined (Fig.1g). How does the calculated shift relate to the observed red-shift in the TA experiments?
4. Can the authors comment on the nature of the bandgaps of these materials? In the above-mentioned report (by the same group), the 1D material has a predicted indirect bandgap with a direct transition slightly higher in energy. What about the 2D-analogue? Could the indirect-direct nature of the bandgap explain the red-shift in the TA experiments? If there's indeed a direct-indirect bandgap, the authors might consider using different pump pulses to get more insights into the charge carrier dynamics of these materials.
5. Can the authors comment on the physical nature of the 4 states they use to fit their TA data? More septically, on page 7 the authors mention for the 1D sample: "The two longest components have similar spectra suggesting no further change in state character". Maybe additional ns-TA experiments can be performed to get more insights into the decay pathway of the long-lived 4th component?
6. The temperature dependent FWHM shows a linewidth increase at low temperatures (<150K) for the 2D sample (Fig. 1c), which is not immediately evident from the spectra plotted in the SI (Figs. S6a,b). Can the authors comment on the non-linear trend of the PL broadening/narrowing observed for the 2D sample compared to other materials reported in the literature, e.g. Wright et al. J. Phys. Chem. Lett. 2021, 12, 3352-3360 or VanOrman et al. Adv. Mater. 2025, 24 19679, Du et al. ACS Nano 2020, 14, 5806-58817]
7. In figure 1 please adjust panel a as it is impossible to observe the 502/588 nm absorption peaks. Please plot the outline of the probe spectrum in the background (or in the SI). Moreover, the labeling of the y-axis is incorrect. The normalized absorbance spectrum and probe/pump PL spectra do not share the same y-axis.

Version 1:

Reviewer comments:

Reviewer #1

(Remarks to the Author)

The authors responded with great detail and additional data and findings to the reviewer comments, which is much appreciated!

This reviewer raised concerns about the novelty, as did other reviewers. It seems the main effects and explanations are mostly expected from what is known from the literature. The response by the authors therefore did not fully rebuttal the concerns about novelty.

Regarding previous technical questions, the authors are now claiming that 5ps and 20ps MD provide very similar VDOS, which would be surprising. Comments:

1. can the authors show a comparison of 5ps and 20ps VDOS data?
2. the MD-VDOS data in Figure S33 appear very broad already at low temperature. Why is this the case? Are there experimental data or previous computational data from literature to compare to? Is this behavior expected?
3. the authors should also specify how the VDOS was calculated. If a formula was implemented, the formula should be specified. If other software was used, it should be cited.
4. the effect of SOC was found to be around 0.2 eV for the band gap. Is this consistent with previous theoretical work for this or related materials?

Reviewer #2

(Remarks to the Author)

Thank you for the detailed responses and thorough revision of the manuscript. Before acceptance, however, I have one main concern that arises from the reply to point 5). The explanation provided for the 502 nm resonance does not appear to be supported by available evidence or theory.

The authors attribute the peak at ~502 nm in the absorption spectrum to "higher-lying excitonic state(s), predominantly 2s but even higher excitons together." This assignment requires more rigorous justification. Does for example the energy spacing between the two resonances match the expected 1s–2s energy difference in the 2D sample? More critically, what mechanism could account for a 2s exciton resonance of comparable or even larger intensity

than 1s? In a Wannier-like exciton picture, 2s is expected to exhibit a substantially weaker oscillator strength than 1s. In halide perovskites, higher-lying excitons, including 2s, have been hardly observed in optical spectra, even at low temperatures [<https://doi.org/10.1038/s41467-018-04659-x>].

Reviewer #3

(Remarks to the Author)

I am pleased with the additional explanations and data sets provided by the authors in this revised manuscript version. Aside from my initial and still standing remark on somewhat lacking novelty, I am happy to recommend publication of this manuscript in its current form in terms of scientific quality.

Version 2:

Reviewer comments:

Reviewer #1

(Remarks to the Author)

The authors provided answers to my comments. The comment about novelty helped to better understand the relevance of the work. The response by the authors to my previous technical comments are not satisfactory:

1. the provided data clearly shows that the VDOS with 5ps trajectory lengths are not converged at all! Textbook knowledge on molecular dynamics provides a rule of thumb that every vibration needs to be sampled at least ten times before it is properly resolved. With vibrational signatures in range of 1 THz or even less in the systems that are studied, this means that the simulation length must be 10 ps or even few times larger. This is seen in the authors' own data, where several new peaks emerge for longer MD runs. If the authors would properly smoothen then 20 ps data, the two curves would not look the same. The authors need to provide further convergence studies (up to 30 or 40 ps, it is the standard in this field) and properly analyze the statistical sampling, in order to finally report technically correct data.

2. Figure 2a in the reference provided by the authors shows a Raman spectrum of MAPbI₃ at 100K. The lowest-energy peaks are not broad at all, one can estimate that the FWHM is around 10cm⁻¹ or even less. There is a clear progression of individual peaks, whilst in the current study there are rather broad features, especially at around 100cm⁻¹. Furthermore, the authors' response is insufficient in explaining the origin of it. 77K means only few meV of thermal energy, what evidence is there of anharmonic effects at this low temperature in these materials? Rather, one might hypothesize that the broadening seen is not physical and an error arising from poor sampling: especially at lower T one must take great care regarding statistical sampling and convergence of MD trajectory (see comment 1). One already sees that the noisy and broad 5ps data at a temperature of 77K becomes sharper and more resolved in the 20ps trajectory. This needs to be investigated further, again the data should be technically correct.

3. Equation 1 provided in the reply is incorrect. C(t) needs to be calculated as mass-weighted velocity autocorrelation function. See Eq. 54 in one of the cited references (Computer Physics Communications 267 (2021) 108033). This is important because without proper mass-weighting the spectrum will be distorted. The authors should implement the correct formula and recalculate all VDOS data.

Reviewer #2

(Remarks to the Author)

The authors have addressed the previous concerns. I am happy to recommend the revised manuscript for publication.

Version 3:

Reviewer comments:

Reviewer #1

(Remarks to the Author)

The authors have successfully addressed the comments by the reviewer and there are no further comments.

Dear Editor,

We thank you and the reviewers for their careful reading of the manuscript (NCOMMS-25-33234-T) and for the feedback that has helped us significantly strengthen our work. We have thoroughly revised the manuscript. We have added new experimental results and extended the calculations. Below we offer answers to reviewers' comments point-by-point and explain the changes made to the manuscript. Besides, in the submitted revision, all changes are marked.

Sincerely, on behalf of the authors,

Tõnu Pullerits

REVIEWER COMMENTS

To improve clarity and avoid repetition, references mentioned in the responses are indicated by bracketed numbers (e.g., Ref. [1], [2]). A full reference list is provided at the end of this response letter. We point out that the manuscript has its own numbering.

Reviewer #1 (Remarks to the Author):

The study by He et al. reports very interesting results on dimensionality-dependence of excitonic effects in 1D and 2D metal halide perovskites. The findings are very interesting and the manuscript well written and presented. It seems to be a good fit for the journal,

We thank the reviewer for the very positive overall judgement.

the following major comments require further clarifications:

1. from the manuscript it was not entirely apparent or discussed in depth why dimensionality would affect exciton localization. For a single electron, the situation is quite clear in the framework of Anderson localization. For excitons this might be similar. Is this the reasoning? This should be discussed in more detail in the manuscript and the novelty aspect of the study should be made more clear in this context.

Answer: We fully agree that the role of dimensionality for the exciton localization needs further clarification in the manuscript. We also agree that the Anderson localization is an important part of the explanation. While originally formulated for single electrons, the general concept of disorder-induced localization is valid for quasi-particles like excitons too, see for example, the discussion about disorder-related exciton localization in light-harvesting complexes (Ref. [1]). From this we know that the disorder leads to localization of the exciton wavefunction, and as known from Anderson localization (Ref. [2-3]), the effect is particularly pronounced in the 1D system. It is also known that electron-phonon interaction in the

localized system is higher compared to the delocalized case (see for example, Ref. [4]). Furthermore, compared to the 2D system, the 1D structure exhibits reduced inter-octahedral connectivity leading to a softer lattice. This means not only higher disorder but also stronger response to the localization in terms of enhanced exciton–phonon coupling. This enhanced coupling drives the exciton self-trapping in the studied 1D system. In the revised manuscript (page 10), we have expanded the following discussion:

“Given the previous results, we noted that the exciton-phonon interaction of the 2D system is drastically smaller than that of the 1D system – the lattice reorganization energy is > 30 times lower. Such a significant difference can be attributed to enhanced disorder-induced Anderson localization in the 1D system [1-4]. The effect is further enhanced by the inherently larger disorder and softness of the 1D system, the former enhancing the localization, the latter making the localization response, as the lattice deformation, stronger.”

The following sentence was added to the abstract:

“The difference originates from enhanced Anderson localization in the 1D system.”

We have also clarified the novelty by highlighting this point, see the following expanded discussion (page 15):

“The role of organic cations in modulating lattice distortions and exciton–phonon coupling is well established in lead-based 2D perovskites [5-17]. Extending this framework, our study shows that even without changing the chemical properties of the organic cations, tuning the dimensionality from 2D to 1D via ligand concentration control leads to pronounced differences in exciton localization and phonon coupling. While the 2D system exhibits robust excitonic features and FE emission consistent with quantum and dielectric confinement [18], the 1D system shows much stronger exciton localization resulting in pronounced STE emission.”

2. The theoretical simulations using AIMD method were not appropriately performed. The authors stated that a 5ps run was used, yet their VDOS (and the experiments) clearly show that dynamics slower than 1ps are present. It is recommended that the authors demonstrate convergence of the VDOS using at least 20 ps of runtime, which is what this reviewer would expect is required to obtain statistically converged data. Moreover, what functional was used during the MD run, and did the authors test the effect of dispersion corrections? These corrections are of key importance in these systems and may significantly impact the reported data. Furthermore, when band gaps were calculated, did the authors consider spin-orbit effects and, if not, how is this justified in these materials? Finally, in the phonon calculations did the authors observe any imaginary modes? These materials are prone to show imaginary modes because there are phase transitions: which structure has been used in these calculations, is it the ground-state structure and were there any imaginary modes present?

Answer: We agree that accurate theoretical treatment is essential, particularly given the complexity and sensitivity of low-dimensional metal halide systems. In the revised manuscript,

we have extended the AIMD simulations to 20 ps at different temperatures using the PBE functional with Grimme's D3 dispersion corrections and SOC (NVT ensemble, 1 fs time step). The VDOS has been recalculated using this extended trajectory, see Supplementary Fig. S33. The overall VDOS profile remains qualitatively similar, particularly in the low-frequency region ($< 150 \text{ cm}^{-1}$), which means this simulation is well converged and consistent with our original 5 ps data. However, the inclusion of dispersion interactions leads to a slight blue shift of the low-frequency peaks, which in fact improves the agreement with experimentally observed phonon modes. This result confirms that dispersion plays a non-negligible role in lattice dynamics but does not qualitatively change the phonon modes relevant to our discussion. We also repeated the bandgap calculations along the MD trajectory using 1 ps time step to investigate possible slower vibrational dynamics (Here we also use the PBE functional with Grimme's D3 dispersion corrections and SOC). The results are presented as Supplementary Fig. S8. The variance of the bandgaps along the 20 ps MD trajectory with 1 ps step is very similar to the variance obtained in our initial calculation of 2 ps trajectory with a shorter step, validating the earlier result. Since the bandgap calculations are quite demanding, we did not repeat them with a short timestep.

We have included the new results in the revised Supporting Information (Supplementary Fig. S8, S33) and updated the Method section in the revised manuscript (page 22) accordingly.

Supplementary Fig. S33 The vibrational density of states (VDOS) and velocity autocorrelation function (VACF) of the 1D system. 77K: (a) and (d); 200K: (b) and (e); 298K: (c) and (f).

Supplementary Fig. S38 Band-gap calculation with two different MD trajectories. Blue points are the same as in Fig. 1g calculated with PBE functional, the orange points are from a 20 ps MD trajectory calculated with Grimme's D3 dispersion corrections and SOC. The root mean square deviation of the calculated band gaps are 160 meV and 210 meV for the blue and orange points, respectively.

To evaluate the influence of spin-orbit effects on the computed band gap, we plotted the total DOS considering SOC and without SOC (Supplementary Fig. S34). The results reveal only marginal differences in the electronic structure, with the maximum deviation in the band gap being approximately 0.15 eV. Given the limited impact of SOC on the band gap, SOC was omitted in the majority of calculations presented in this work. All reported calculations were performed using locally optimized structures corresponding to local minima on the potential energy surface (PES). The phonon band structure (Fig. R1) of the investigated cell exhibited no imaginary frequencies, aside from negligible values attributable to numerical artifacts. This confirms the dynamical stability of the structure and supports its identification as a local minimum on the PES. The explanation of the SOC effect and corresponding figures have been added to the revised manuscript (page 22, Methods) and Supporting Information (Supplementary Fig. S34).

Supplementary Fig. S34 Total DOS using PBE functional, considering SOC and without SOC. The results showed only minor differences in the electronic structure (the maximum deviation

in the band gap is approximately 0.15 eV); thus, for consistency, the band structure calculations were performed using the non-spin-polarized approach.

Comment Fig. R1 Phonon band structure of ODASn₂I₆ with PBE level of theory.

Reviewer #2 (Remarks to the Author):

The manuscript by Y. He et al reports a study of 1D and 2D tin-halide perovskites addressing in particular exciton-phonon couplings by photoluminescence and transient absorption spectroscopy. Emission from 2D samples is found to be dominated by free excitons, whereas in the 1D case self-trapped excitons dominate. Exciton-phonon coupling is observed to be ~ 2.5 times larger in 1D compared to the 2D structure. Phonon modes are assigned based on DFT calculations (VASP). The manuscript is generally well written.

We agree with the general summary of our work and thank the reviewer for the positive overall judgement of the manuscript.

Both optical and electronic properties of halide perovskites are generally strongly influenced by the interaction with the polar lattice. Additionally, the presence of organic spacers in 2D systems gives rise to unique and complex optical properties, so far not well understood yet.

Tin halides represent a promising lead-free material platform and are thus highly interesting for applications although they are known to be more prone to stability issues. Importantly, low dimensional tin halide perovskites are significantly less thoroughly investigated than lead-based counterparts. As such this study reports a highly relevant and timely topic and can contribute an important step in filling the gap of knowledge on optical excitations and lattice behavior of low-dimensional tin-halide perovskites.

We agree with the reviewer that tin halides represent an important lead-free material platform and we thank her/him for the good words.

Despite the relevance of the work, I have a few concerns and questions (reported below) that need to be addressed before a decision on publication can be made.

We thank the reviewer for the constructive comments/questions and we provide our answers below.

1) One of the main findings of the manuscript is that the 1D system shows much larger exciton-phonon coupling than the 2D system for the dominantly coupled mode. While this result is plausible due to stronger exciton localization in 1D, the manuscript currently lacks a detailed physical explanation for this finding.

Answer: The observed differences between the 1D and 2D systems can be related to Anderson localization in disordered systems (Ref. [1-4]). We note that in the 1D system, the four corner-sharing $[\text{SnI}_5]^{3-}$ moieties align along the c-axis like a glide plane separated by ligands, leading to confinement of electronic wavefunctions along one dimension. Such 1D tin-halide framework is structurally soft and prone to lattice distortion due to the lower dimensional connectivity and reduced steric constraints compared to the 2D layered structure. The distortions and dimensionality cause strong localization (Anderson localization is significantly more efficient in 1D than in 2D system) that enhances exciton-phonon interaction. We have now expanded the relevant discussion (page 10, revised manuscript) to clarify the underlying mechanism:

“Given the previous results, we noted that the exciton-phonon interaction of the 2D system is drastically smaller than that of the 1D system – the lattice reorganization energy is > 30 times lower. Such a significant difference can be attributed to enhanced disorder-induced Anderson localization in the 1D system [1-4]. The effect is further enhanced by the inherently larger disorder and softness of the 1D system, the former enhancing the localization, the latter making the localization response, as the lattice deformation, stronger.”

2) The role of the organic spacer in modulating inorganic lattice distortion, and consequently the exciton-phonon interaction having considerable impact on optical properties, has been previously proposed for 2D perovskites (e.g. <https://doi.org/10.1016/j.trechm.2019.04.003>). The present manuscript supports that these mechanisms may be analogous in tin-based systems.

Answer: We agree with the reviewer that organic spacers play a key role in modulating exciton-phonon coupling and the resulting optical properties in 2D lead halide perovskites. In the revised manuscript, we have added discussion of this point together with the reference (Ref. [18]) provided. Our study extends this framework to low-dimensional tin halides by systematically varying the organic ligand concentration to tune structural dimensionality from 2D to 1D. This dimensional reduction enhances exciton localization and then exciton-phonon

coupling due to reduced inter-octahedral connectivity and increased lattice softness. We now clarify that ligand-induced excitonic and lattice dynamics can be achieved not only by changing the ligand identity but also through compositional control over organic cations (page 15, revised manuscript):

“The role of organic cations in modulating lattice distortions and exciton–phonon coupling is well established in lead-based 2D perovskites [5-17]. Extending this framework, our study shows that even without changing the chemical properties of the organic cations, tuning the dimensionality from 2D to 1D via ligand concentration control leads to pronounced differences in exciton localization and phonon coupling. While the 2D system exhibits robust excitonic features and FE emission consistent with quantum and dielectric confinement [18], the 1D system shows much stronger exciton localization resulting in pronounced STE emission...”

“In summary, our combined experimental and computational analysis supports and extends the established polaronic framework of 2D lead halides. The tin halide systems investigated here exhibit distinct excitonic and vibrational dynamics, underscoring the critical role of dimensionality-dependent exciton–phonon interactions in modulating both lattice dynamics and electronic transitions, such as exciton self-trapping. This modulation, achieved only through concentration-dependent tuning of organic components, enriches the previous polaronic framework and triggers the broader applicability of ligand-mediated exciton-phonon coupling to lead-free hybrid metal halides.”

3) When exciting at 400 nm, likely free charge carriers, rather than bound excitons, are predominantly initially generated. How do the authors ensure that the vibrational modulations observed in the nonlinear TA signal originate from exciton-phonon rather than charge-phonon coupling?

Answer: We thank the reviewer for raising this important point. We have observed a large Stokes shift and broad emission in the 1D system, which indicates the formation of STE. For this material, the very large exciton binding energy can be expected by referring to the other well-studied STE-based low-dimensional halide perovskites (Ref. [19-20]). Therefore, under the band-edge excitation of 400 nm, the observed oscillation signals in the 1D system should originate from the strong exciton-phonon coupling rather than the charge-phonon coupling.

In the 2D system, we agree that such high-energy excitation can generate hot carriers; thus, charge–phonon coupling should be taken into consideration. The hot carrier cooling process with a lifetime of 210 fs has been evidenced in the excitation-intensity-dependent TA spectra. In this case, we would expect that there may be an overlap of charge-phonon coupling (< 210 fs) and exciton-phonon coupling (> 210 fs). However, the excited-state oscillations in the 2D system are generated via the displacive mechanism, which means the wavepacket starts on the displaced excited state potential surface at a considerable distance from the equilibrium position. As shown in Fig.4a, the phonon period of the dominant vibrational mode at 47 cm^{-1} is $\sim 1.4\text{ ps}$, which is much slower compared to hot carrier cooling. Then, most of the observed

oscillation signals in the 2D system should be induced due to the exciton-phonon coupling. Additionally, we have observed the well-defined excitonic absorption peaks in the 2D system (502 nm: higher-lying excitonic state; 588 nm: band-edge excitonic state). These features could also support the pronounced role of exciton–phonon coupling in this system.

We have now expanded the relevant discussion in the revised manuscript (page 9):

“We found that at 77 K oscillations appear even in the 2D system. Since in this case we excite high above the lowest band, most likely creating free charges, the oscillations may be partially driven by the charge carrier coupling to the phonons. However, the carrier cooling happens with a time constant of 210 fs, which is much faster than the phonon oscillation period. Therefore we conclude that the exciton-phonon coupling dominates the phonon coherence in the 2D system.”

4) In the TA measurements of the 2D sample, two GSB and three ESA bands are observed. The observed red-shifts of the GSB bands are explained in terms of initial carrier cooling and excitation dependent effects such as EIS. The origin of the ESA bands in contrast is not clearly discussed in the manuscript. Moreover, lead-halide perovskite have been reported to be sensitive to both excitation induced shift and excitation induced dephasing (<https://doi.org/10.1021/acs.jpcc.5b00148> ; <https://doi.org/10.1146/annurev-physchem-102822-100922>; <https://doi.org/10.1021/acs.jpcc.2c00658>). EID arises from pump-induced broadening of the exciton resonance, that can be due to exciton-charge carrier or exciton-exciton interactions at increasing high fluence. These can result in typical lineshape distortions and appearance of side bands in nonlinear experiments, such as TA, and have been widely investigated both experimentally and theoretically in conventional semiconductors. Can the authors exclude here that the lineshapes are not affected by EID? Since experiments at different excitation density are performed, an excitation-density-dependent lineshape analysis could help to check this point. Additionally, it should be ensured that the experiments are performed at carrier densities below the Mott transition (<https://doi.org/10.1038/s41467-020-14683-5>; <https://doi.org/10.1021/acs.jpcclett.9b01936>).

Answer: We thank the reviewer for pointing out this possibility. The ESA bands immediately appear next to the two GSB bands. The ESA bands are likely associated with intra-band transitions but they can also be partially due to the exciton-induced shift (EIS) of the bands (two distinct excitonic absorption peaks in the steady-state spectrum, Fig. 1a) and exciton-induced dephasing (EID). To evaluate the possible EID effect for the observed TA signal, we carefully analyzed spectral lineshapes and their widths as a function of pump fluence and population time. Within the applied pump fluence range, we observe that the shape of TA spectra remains largely unchanged (Fig. R2), showing minor changes (< 15 meV) in the linewidth of the GSB peak at 585 nm (Supplementary Fig. S10). Additionally, the linewidth of this GSB peak remains relatively constant over the probed population time. The ESA band tail extends from 650 nm to 900 nm when the excitation fluences increase. In principle, this could be the effect of EID but more likely corresponds to the exciton-related band renormalization

(EIS). Of course, we cannot exclude EID contribution. However, our intensity-dependent measurements are performed at room temperature (the references provided by the reviewer about EID are at cryogenic temperatures), thereby the dephasing is already very fast (shorter than a few tens of fs) and the additional broadening due to the EID would be difficult to identify.

We have now extended the discussion of the origin of the ESA bands in the revised manuscript (page 6):

“Three excited state absorption (ESA) bands are observed together with the two GSB bands. The ESA bands can have contributions from intra-band transitions, exciton-induced bandshifts (EIS) [21] and exciton-induced dephasing (EID) [22-24]. Partial overlap of the ESA bands with the GSB makes it difficult to disentangle the corresponding contributions. (Supplementary Fig. S10.)”

Comment Fig. R2 The fastest components (τ_1) are extracted from excitation intensity-dependent evolution-associated spectra (EAS) for comparison. n is the excitation density with the unit of excitation/cm³/pulse.

Supplementary Fig. S10 The line width of the GSB signal probed at 585 nm from the lineshape analysis of the TA spectrum at the time delay of 200 fs. The pump fluence ranges are (a) f_1 : 3.4×10^{13} photon/cm²/pulse; (b) f_2 : 3.4×10^{14} photon/cm²/pulse; (c) f_3 : 1.7×10^{15} photon/cm²/pulse.

Regarding the Mott transition, in a 2D spectroscopy study of bulk perovskite crystals (Ref. [25]) a blueshift of the zero crossing of the spectra with increasing the excitation concentration above 10^{18} cm^{-3} was taken as evidence for the Mott transition. Here we have used significantly higher excitation concentrations but the zero crossing does not show any blue shift. Quite the opposite, we observe a red-shift with increasing the excitation intensity, see Supplementary Fig. S11. We take this as evidence that our experiments are performed below the Mott transition. The exciton binding energy in bulk perovskite studied in [ref] is much lower than in the 2D and 1D systems studied here. Therefore in our systems the Mott transition occurs at significantly higher excitation intensities than what is used in our experiments.

The following text was added to the Supplementary Information Note 3 together with Figure S11.

“In the intensity-dependent measurements, the excitation concentration is controlled to be well below the threshold of Mott transition, which is supported by the absence of blueshift around the zero-crossing in the intensity-dependent spectra (Supplementary Fig. S11) [25].”

Supplementary Fig. S11 Normalized ΔA spectra. No blue shift of the zero-crossing is observed with increasing intensity, showing that the photocarrier concentration stays below the Mott-transition. n is the excitation density with the unit of excitation/ cm^3 /pulse.

5) For the 2D system presented in this manuscript, two distinct excitonic peaks are observed in the absorption spectrum at 502 nm and 588 nm. The physical origin of these transitions should be addressed. Are these related to different exciton species, or polaronic states?

Answer: These two bands are of excitonic origin. We assign the lower-energy peak ($\sim 588 \text{ nm}$) to the lowest excitonic transition (1s exciton) arising from the well-known quantum and dielectric confinement effects in the 2D layered metal halides (Ref. [18]). This state is strongly confined within the inorganic layers. The higher-energy peak ($\sim 502 \text{ nm}$) is attributed to a higher-lying excitonic state(s), predominantly the 2s but even higher excitons together with the band edge absorption contribute at the blue side. Similar multiple excitonic resonances have been reported in related 2D perovskites due to the presence of numerous electronic transitions (Ref. [26-27]). The exciton-phonon interaction in the 2D system is not strong and we did not observe evidence for polaronic features in the absorption nor emission spectra.

Instead, the low-energy emission tail is attributed to the surface trap states. The expanded explanation has been added to the revised manuscript to improve clarity (page 4):

“In the absorption spectrum, we observed the excitonic peaks at 502/588 nm for the 2D system and 359 nm for the 1D system. The low-energy peak (588 nm) in the 2D system corresponds to the lowest exciton transition 1s. The 502 nm band is dominated by the 2s transition but most likely includes even higher exciton transitions and the band edge absorption.”

6) The manuscript does not mention the excitonic structure or binding energy for the 2D and 1D systems investigated in this work. 2D perovskites exhibit pronounced exciton fine-structure splitting of band-edge excitons with bright-bright splitting of a few meV, dark-bright splittings around 10 meV, and in some Br-based materials even tens of meV. Fine structure appears important in general for light-matter interaction of these quasiparticles (DOI: 10.1126/sciadv.abk0904; <https://doi.org/10.1063/5.0048490>; <https://doi.org/10.1021/jacs.3c11957>;). Estimating exciton binding energies and discuss the excitonic structure of these materials would enrich the analysis.

Answer: We thank the reviewer for highlighting the importance of excitonic fine structure and binding energy in low-dimensional perovskite systems. Here in our 2D system with a layered structure, the lower exciton bands originate from the Rydberg series of the Wannier-Mott exciton (1s and 2s exciton states are at 2.11 eV and 2.47 eV, respectively), allowing us to estimate the exciton binding energy of approximately 400 meV by using the classic 2D hydrogen Rydberg series with energies $E_{Ns} = E_G - R_y / (N - 1/2)^2$, yielding the R_y (Rydberg energy) of ~ 0.104 eV and the E_G of ~ 2.516 eV (Ref. [28]). This result is consistent with the reported values from closely related 2D tin halide perovskites such as (PEA)₂SnI₄, (OA)₂SnI₄, (BA)₂SnI₄ (Ref. [26-27, 29-30]). Large binding energy is the result of strong dielectric and quantum confinement in 2D layered structures, supporting the stability of excitons at room temperature. For the 1D system, which exhibits STE emission, strong exciton localization can lead to even higher exciton binding energy, in line with other STE-based metal halides (Ref. [19, 31-32]). This behavior is typically associated with enhanced lattice distortion and strong exciton-phonon coupling in low-dimensional structures.

The fine-structure splittings due to spin-orbit coupling, exchange interactions, and lattice symmetry breaking, lead to bright-dark exciton manifolds separated by a few to tens of meV (Ref. [27, 33-34]). These effects are hidden by the relatively broad spectral bands and are not targeted in our study. The 1D tin halide system likely exhibits even stronger exciton localization and modified fine structure due to enhanced lattice distortion and reduced symmetry. These effects could influence the exciton-phonon coupling strength and warrant future detailed spectroscopic investigations.

The following discussion has now been added to the Supporting Information (page 4):

“The exciton binding energy in the 2D system is estimated to be approximately 400 meV, by using the classic 2D hydrogen Rydberg series with energies $E_{Ns} = E_G - R_y/(N - 1/2)^2$, where the 1s and 2s exciton states are at 2.11 eV and 2.47 eV, respectively. This yields the R_y (Rydberg energy) of ~ 0.104 eV and the E_G of ~ 2.516 eV [28]. This result is consistent with the reported values from closely related 2D tin halide perovskites such as $(PEA)_2SnI_4$, $(OA)_2SnI_4$, $(BA)_2SnI_4$ [26-27, 29-30]. The large binding energy reflects the strong dielectric confinement and reduced dimensionality in the 2D system. The clear excitonic features observed in the absorption and photoluminescence spectra are consistent with the presence of tightly bound excitons that are stable at room temperature. In the 1D system, which shows characteristic STE emission, the stronger exciton localization is expected due to enhanced lattice distortion and electron–phonon coupling. Such localization is typically associated with even larger exciton binding energies, as observed in other STE-emissive 1D or low-symmetry halide systems [19, 31-32]. While our current measurements do not allow direct quantification of the binding energy, the spectral characteristics suggest that exciton confinement and localization are significant in both systems.”

7) As a general remark, I strongly recommend to put the discussion in the context of the large body of literature that is available for low dimensional lead-halide counterparts highlighting potential similarities and differences with the present findings on tin-based 2D and 1D structures.

Answer: We thank the reviewer for this suggestion. We have thoroughly revised the ‘Discussion’ section by referring to 13 related articles [5-17] together with an extended comparison of our findings, focusing on the similarities in mechanism, differences in the impact of dimensionality on light-matter interactions, including exciton self-trapping and distinct phonon modes, which aims at drawing parallels and contrasts between tin- and lead-based systems. The expanded discussion is shown below (revised manuscript, page 15):

“The role of organic cations in modulating lattice distortions and exciton–phonon coupling is well established in lead-based 2D perovskites [5-17]. Extending this framework, our study shows that even without changing the chemical properties of the organic cations, tuning the dimensionality from 2D to 1D via ligand concentration control leads to pronounced differences in exciton localization and phonon coupling. While the 2D system exhibits robust excitonic features and FE emission consistent with quantum and dielectric confinement [18], the 1D system shows much stronger exciton localization resulting in pronounced STE emission. The excitation-intensity dependent TA measurements reveal that the STE population dynamics in 1D structure remain largely unaffected by changes in excitation conditions, however, the FE dynamics are significantly influenced by hot carrier cooling and Auger recombination, particularly at higher excitation intensities. The temperature-dependent TA measurements further demonstrate that the coherent vibrational wavepackets on STE state in the 1D system are dominated by a phonon mode at 106 cm^{-1} involved in the wagging and asymmetric

stretching vibration in [SnI₅], while the coherent vibrational wavepackets on FE state and the ground state in the 2D system have significantly lower frequencies below 50 cm⁻¹.

In summary, our combined experimental and computational analysis supports and extends the established polaronic framework of 2D lead halides. The tin halide systems investigated here exhibit distinct excitonic and vibrational dynamics, underscoring the critical role of dimensionality-dependent exciton–phonon interactions in modulating both lattice dynamics and electronic transitions, such as exciton self-trapping. This modulation, achieved only through concentration-dependent tuning of organic components, enriches the previous polaronic framework and triggers the broader applicability of ligand-mediated exciton–phonon coupling to lead-free hybrid metal halides.”

Minor comments:

Fluences are reported without units in Fig. 2 and related Supplementary Figures.

Answer: We have now added the units for the pump fluences (photon/cm²/pulse) to Fig. 2 and all related Supplementary Figures for clarity. The captions are also revised to explain the units of both excitation fluence and intensity.

With respect to vibrational dynamics, I recommend showing residual maps in the manuscript, e.g. as additional panels in Fig 3, not only in the SI, since these maps are essential for the results.

Answer: The residual maps have been added as panels b and e in Fig. 3.

In Fig. 3 panel c and d the labels for y-axes (Frequency (cm⁻¹) ?) are missing.

Answer: The missing y-axis labels for Frequency (cm⁻¹) have been added.

Reviewer #3 (Remarks to the Author):

The work by He et al. reports on the role of the sample dimensionality on the exciton–phonon interactions in thin-layered Sn-halide semiconductors. Previous work by (part of) the authors reported on a self-trapped exciton for ODASn₂I₆, which is absent for the 2D counterpart ODASnI₄. With the use of ultrafast transient absorption spectroscopy, the dimensionality-dependent charge carrier dynamics are studied as a function of temperature to elucidate the lattice dynamics in low-dimensional tin-based semiconductors. By taking the fast Fourier transform of the lower-temperature fs-TA data, I-Sn-I vibrational modes are identified, which are used to rationalize charge carrier dynamics observed for both 1D and 2D systems.

Overall, I have some issues with the novelty of the work. A large part of the steady-state and transient spectroscopy experiments (Figs. 1a,d,f and Figs. 2d–f, Fig. 4c) appear to be reproduced from earlier work by the group [Adv. Optical Mater. 2025, 13, 2402061].

Additionally, the authors conclude that the free exciton dynamics in 2D thin films are significantly influenced by hot carrier cooling and higher-order recombination, which is well established for inorganic-organic semiconductors. This is reducing the novelty of this work to the detailed analysis of the vibrational modes extracted from the low-temperature TA experiments. As such, I unfortunately cannot recommend this work for a high-impact broad audience outlet like Nature Communications and would suggest publication of this additional (but nice) data set in a more specialized outlet.

Answer: We thank the reviewer for careful consideration and for general appreciation of our work (“the data are nice!”). Indeed, the temperature dependence of the emission of the same 1D system as here has been published before. However, the overlap with the current comparative study of the exciton-phonon interaction in 1D and 2D systems is minimal. It is limited to the absorption and emission spectra of one compound (Fig. 1a and d compare the spectra of 1D and 2D, the former spectra appear also in our earlier work). Such spectra constitute a basic and necessary characterization of material in any publication. Figure 1f, in contrast, extends the temperature range of the analysis by more than a factor of two, which constitutes a clear advance beyond our earlier work enabling a new level of precision in determining the exciton–phonon coupling not possible in our previous work.

The transient absorption data in Fig. 2d employ a similar technique (like the earlier work), but the purpose here is distinct. These data highlight the pronounced differences compared to the 2D case (Fig. 2a). Moreover, the intensity-dependent kinetics shown in Figs. 2e and 2f are new additions that provide valuable information about nonlinear carrier relaxation dynamics and the differences between the two systems, which were not investigated previously.

The schematic in Fig. 4c has been entirely reworked to reflect a substantially refined model illustrating the processes in the 1D system, with a focus on the early-time dynamics derived from our current experimental data. In contrast, the earlier schematic focused on explaining the temperature dependence of the luminescence, thereby focusing on long timescales and the quasi-equilibrium conditions. Thus, this figure represents a conceptual advancement, not a reproduction.

We respectfully acknowledge the reviewer’s point regarding the established nature of hot carrier cooling and higher-order recombination in hybrid semiconductors. These processes form a minor sideline of the work – they are visible in the data and we point it out. We agree, these are not the novel aspects of our work. The novelty of our work lies in elucidating how the dimensionality, specifically changing from 1D to 2D systems, influences both the electronic and vibrational dynamics in low-dimensional tin-halide perovskites, a direction nobody has addressed (including us) before. Unlike previous studies that focus on typical lead-based perovskites, our work explores tin-based halides, which are significantly less studied yet crucial for environmentally friendly optoelectronics.

Taken together, we believe the combined expansion of experimental scope, comparison across different dimensionalities, and detailed analysis of excitonic/vibrational dynamics establishes a clear advance beyond existing literature including our own prior work. These insights bridge the link between the structure and properties in tin halides, offering an important step forward in understanding structure-property relationships in this emerging material family.

Specific comments:

1. Can the authors comment on the self-absorption (if any) for ODASnI₄? The asymmetric shape of the PL spectrum suggests there is a considerable self-absorption. As panel a suggests a considerable overlap of the PL and absorption spectra, how does this affect the determined FWHM used to determine the Huang–Rhys (S) factor?

Answer: We thank the reviewer for this important comment. Indeed, in 2D system absorption and PL overlap and some self-absorption are likely to happen. We plot the absorption and PL spectrum together (See Fig. R3). If significant self-absorption takes place, we would expect that features opposite to the absorption spectrum appear in PL – for example, a shoulder or dip-like feature at around 550 nm where the absorption has a clear minimum. We do not see any hint of such a dip. Therefore, we conclude that self-absorption is negligible. However, we agree that some self-absorption is likely to take place in ODASnI₄. The following text was added to the revised manuscript (page 5):

“In the 2D system, the overlap with the absorption band might lead to partial self-absorption, which could result in an underestimation of the FWHM. Additionally, the weak emission tail at 660 nm, coming from the surface trap states (SUT), makes the line-width analysis for the S factor unreliable (Supplementary Fig. S9).”

This said, we would like to also clarify that the FWHM of the PL spectrum in the 2D system was used solely for qualitative comparison. The PL spectrum is broadened by the emission from surface trap states. These effects add to the spectral broadening and make it challenging to extract reliable information about the electron–phonon coupling from the FWHM alone. For these reasons, in our work, we did not use PL FWHM to evaluate the Huang–Rhys (S) factor in the 2D system. Instead, we quantified the S factor of the 2D ODASnI₄ using the method derived from the oscillatory components in the fs-TA measurements. Please see the detailed explanation on page 10 in the main text and Supplementary Note 4.

Comment Fig. R3 The steady state absorption and emission spectra are plotted for further illustration. The P1(502 nm) and P2 (588 nm) are the two excitonic peaks. The Stokes shift is also labelled inset.

2. On page 3, second paragraph, it is mentioned the fs-TA experiments were performed under resonant excitation conditions. In the manuscript it is mentioned that the absorption peaks are at 502 nm and 528 nm. However, a 400 nm pump pulse is used. Can the author clarify why they excite so off-resonance at 400 nm?

Answer: The main target of the work is comparative analyses of the 1D and 2D systems. The absorption of the 1D system is in the UV region, while the 2D system absorbs further to the visible region with bands at 502 and 588 nm, see Fig. 1a. To keep the measurement conditions the same in these two materials, we choose 400 nm pump excitation which is resonant with 1D system (see Fig. 1a, blue line) and excites also the 2D system (green line). Though for the latter, it is significantly higher in energy than the lower bands. This means that in the 2D system excitation generates free carriers or hot excited states above the band edge. Thereby ultrafast relaxation dynamics, carrier cooling, and exciton formation processes show up prior to exciton localization at the band-edge states.

We admit that the word “resonant excitation” in the manuscript was imprecise (only refers to the 1D system) and have revised the text to clarify that the pump excitation was 400 nm rather than strictly resonant with the excitonic absorption peaks. We thank the reviewer for pointing out this discrepancy. The clarification has been added in the revised manuscript (page 3).

3. With the use of electronic structure calculations, the time-dependent bandgap energy is determined (Fig.1g). How does the calculated shift relate to the observed red-shift in the TA experiments?

Answer: We thank the reviewer for raising this question. We would like to clarify that the electronic structure calculations presented in Fig. 1g are for the 1D system (ODASn₂I₆), which exhibits self-trapped exciton emission and shows no clear red-shift signature in the TA spectra (See Supplementary Fig. S18 – only excited state absorption is observed in the 1D system). The analysis provides valuable mechanistic insights into how specific phonon modes couple to the electronic structure and modulate the bandgap. From the autocorrelation of the time-

dependent bandgap in the 1D system, we extracted a spectral density function that reveals the vibrational modes responsible for bandgap modulation, along with their corresponding coupling strengths (Huang–Rhys factors). Since the calculations correspond to the electronic ground state nuclear dynamics, it does not explicitly catch the self-trapping and the related shift of the PL. However, the calculations do provide information about the displacement of the potential energy surfaces of the modes and the related reorganization energy strongly supporting the case for the self-trapped excitons and the related shifts of the bands.

4. Can the authors comment on the nature of the bandgaps of these materials? In the above-mentioned report (by the same group), the 1D material has a predicted indirect bandgap with a direct transition slightly higher in energy. What about the 2D-analogue? Could the indirect-direct nature of the bandgap explain the red-shift in the TA experiments? If there's indeed a direct-indirect bandgap, the authors might consider using different pump pulses to get more insights into the charge carrier dynamics of these materials.

Answer: For the 2D system, the absorption spectrum exhibits a sharp excitonic absorption peak at the band edge (Fig. 1a), which is a characteristic signature of an allowed direct transition (Ref. [35-36]). Such sharp peaks are typically absent in indirect bandgap materials. Given the extremely small Stokes shift (~27 nm), we consider that the radiative recombination occurs without significant phonon involvement or momentum change, further supporting a direct bandgap (Ref. [37]). Here, the low PLQY does not arise from an indirect bandgap but stems from non-radiative recombination pathways (surface trap states, exciton dissociation, etc.), which are common even in direct-bandgap materials.

Regarding the observed red shift (~3 nm) in ground-state bleaching bands, it is attributed to the partial compensation between the bandgap renormalization (BGR) and the Burstein-Moss effect (BMS). As shown in Supplementary Fig. S13, bandgap renormalization arises from many-body Coulomb interactions at high photoexcited carrier densities, which lower the energies of the conduction and valence band edges, effectively narrowing the optical bandgap. This results in a transient red-shift of the absorption edge on ultrafast timescales, which is a well-established effect in semiconductors and has been directly observed in both 3D and 2D metal halide perovskites (Ref. [38-39]). In contrast, the Burstein–Moss effect, arising from state filling and Pauli blocking, leads to a blue shift of the absorption onset at high carrier densities (Ref. 40-41]). The balance between these opposing effects depends on the excitation fluence and material-specific screening behavior. In our case, the observed small red-shift suggests that BGR slightly outweighs Burstein–Moss contributions. The corresponding explanations and figures are added in the revised manuscript (page 6), and Supporting Information (page 9):

“We here exclude the possibility of a direct-to-indirect bandgap transition since the 2D system shows a direct bandgap and the GSB red-shift appears promptly (within 100 fs) scaling with excitation density (more details see Supplementary Note 3, Fig. S12).”

“Regarding the observed GSB red-shift in the TA spectra of the 2D system, we first exclude the possibility of a direct-to-indirect bandgap transition. The absorption spectrum exhibits a sharp excitonic absorption peak at the band edge, which is a characteristic signature of an allowed direct transition [35-36]. Given the extremely small Stokes shift, we consider that radiative recombination occurs without significant phonon involvement or momentum change, further supporting a direct bandgap [37]. Here, the low PLQY does not arise from an indirect bandgap but stems from non-radiative recombination pathways (surface trap states, exciton dissociation, etc.), which are common even in direct-bandgap materials.

Secondly, BGR arises from many-body Coulomb interactions at high photoexcited carrier densities, which cause a downward shift in the conduction and valence band edges, effectively narrowing the optical bandgap (Supplementary Fig. S13b). This results in a transient red-shift of the absorption edge on ultrafast timescales, which is a well-established effect in both 3D and 2D metal halide perovskites [38-39]. By contrast, the Burstein–Moss effect (BMS), arising from state filling and Pauli blocking, leads to a blue-shift of the absorption onset at high excitation densities (Supplementary Fig. S13a) [40-41]. The relative contributions of these two competing effects depend on excitation fluence and carrier screening. In the 2D system, we observed a small red-shift, suggesting that BGR slightly outweighs Burstein–Moss contributions.”

We agree that the excitation-energy-dependent charge carrier dynamics are interesting for 2D halide perovskites. In this study we focus on the dimensionality-dependent exciton-phonon coupling in organic-inorganic tin halides. Therefore, the detailed excitation-energy-dependent charge carrier dynamics are beyond the scope of this study and will be targeted in our future work.

Supplementary Fig. S13 (a) Burstein–Moss shift, and (b) Bandgap renormalization in photoexcited hybrid metal halides.

5. Can the authors comment on the physical nature of the 4 states they use to fit their TA data? More septicly, on page 7 the authors mention for the 1D sample: “The two longest components have similar spectra suggesting no further change in state character”. Maybe additional ns-TA experiments can be performed to get more insights into the decay pathway of the long-lived 4th component?

Answer: Indeed, the physical meaning of the kinetic components that emerge from our global analysis of the transient absorption data is important. The reviewer is right, ns and even longer time-scale experiments that we are presenting in the manuscript give important insights about the longer decay pathways. We summarize here an intuitive and explicit summary of the four components identified from the global analysis of the transient absorption data and described in the manuscript:

- (1). First (fastest) component - initial excited state localization, cooling and STE formation:
This component reflects the very early processes right after photoexcitation. In the 1D system, the electronic excitation is initially somewhat delocalized due to the closely packed 1D chains. Because of strong exciton–phonon coupling, this state rapidly loses coherence and becomes localized, similar to excitonic polaron formation. At the same time, excess energy is lost to lattice vibrations (cooling). We therefore assign this component to the combined effects of excited-state localization, energy relaxation, and the onset of STE formation. This interpretation is further supported by the spectral similarity between the first two components in the global analysis and the observed blue-shift in the excited-state absorption (ESA) features, consistent with a relaxation to lower energy levels.
- (2). Second component (~ 2.8 ps) - formation of a new STE state (STE_2):
This component reflects the system’s evolution into a new and more stabilized excitonic configuration. In the spectra, we observe that the initial broad ESA feature develops into a two-band structure (Supplementary Fig. S18c), which we attribute to the formation of a distinct self-trapped exciton state (STE_2). This marks the transition from a loosely localized state to a more relaxed, energetically favorable one.
- (3). Third component (~ 47 ps) - nonradiative decay pathway:
The third component exhibits a spectral shape nearly identical to the longest-lived state and has a relatively larger amplitude. Based on the photoluminescence quantum yield of $\sim 37\%$ (Supplementary Fig. S3), we interpret this as a nonradiative decay pathway that quenches about 60% of the excited states. This quenching is likely due to structural inhomogeneities in the material.
- (4). Fourth component ($\gg 5$ ns)-radiative recombination of long-lived STE state:
The longest component represents the remaining excitations that are not affected by the nonradiative channel and decay radiatively. We conducted the ns-TA measurement as

suggested by the reviewer. The TA spectrum shows a broad ESA signal across the whole wavelength detection window (Supplementary Fig. S19). The characteristic decay probed at 583 nm in (b) lasts over 2 μ s, which is consistent with the TRPL measurements (Supplementary Fig. S5b). Clearly, this component is responsible for the observed broad emission in the steady-state measurement.

We have extended the following explanation to further clarify the longest component, which can now be specified to have a time constant of 2 μ s (page 7, revised manuscript). The figures are also added in the Supporting Information (Supplementary Fig. S19).

“The longest component ($\gg 5$ ns) represents the remaining excitations that decay radiatively, consistent with the observed long PL lifetime (Supplementary Fig. S5b). The ns-TA spectrum presents the broad ESA signals across the whole wavelength detection window and the decay probed at 583 nm lasts over 2 μ s, which is well in line with the TRPL measurements (Supplementary Fig. S19). Therefore, we assign this component to the radiative recombination of the long-lived STE state.”

Supplementary Fig. S19. (a) TA spectrum measured at the time window ranging from 0 ns to 2000 ns. (b) The time trace is probed at 583 nm.

6. The temperature-dependent FWHM shows a linewidth increase at low temperatures (<150K) for the 2D sample (Fig. 1c), which is not immediately evident from the spectra plotted in the SI (Figs. S6a,b). Can the authors comment on the non-linear trend of the PL broadening/narrowing observed for the 2D sample compared to other materials reported in the literature, e.g. Wright et al. J. Phys. Chem. Lett. 2021, 12, 3352-3360 or VanOrman et al. Adv. Mater. 2025, 24 19679, Du et al. ACS Nano 2020, 14, 5806-58817]

Answer: We thank the reviewer for calling attention to this subtle yet meaningful observation. The observed non-monotonic increase of the PL FWHM below ~ 150 K in the 2D sample is indeed a subtle feature. This atypical behaviour might be induced by one of the following factors or a combination of them, such as localized disorder, phase heterogeneity, or impurities. Although no sharp phase transition is evident in our temperature-dependent PL spectra, subtle temperature-dependent lattice relaxations or enhanced dynamic disorder may

modulate trap distributions or charge localization, further affecting linewidth and lineshape (Ref. [42-44]). The scattering from ionized impurities can also contribute to the inhomogeneous broadening of the width, like MAPbBr₃ and MAPbI₃ (Ref. [45-46]). Similarly, the 2D material might suffer disturbance from impurities at low temperatures.

To address this abnormal linewidth broadening below 150 K, we have now added a dedicated discussion in Supplementary Note 2, page 3 and an enlarged figure with the labelled FWHMs in the Supplementary Fig. S7:

“We noticed that the observed deviation from the expected monotonic narrowing of the PL linewidth below ~150 K in the 2D sample likely reflects the onset of additional inhomogeneous broadening mechanisms that become relevant at low temperatures (Supplementary Fig. S7). This abnormal behavior could be induced by localized disorder, phase heterogeneity, or the scattering from the impurities at low temperature, which are well-explored in the typical lead halide perovskites, such as MAPbBr₃ and MAPbI₃ [42-46]. While we have consistently observed this behavior across multiple measurements, a more conclusive mechanistic understanding would require further analysis, which lies beyond the scope of the present study.”

Supplementary Fig. S7 The FWHMs of PL spectra for the 2D system measured at 100 K (purple), 125 K (red), and 150 K (black).

7. In figure 1 please adjust panel a as it is impossible to observe the 502/588 nm absorption peaks. Please plot the outline of the probe spectrum in the background (or in the SI). Moreover, the labeling of the y-axis is incorrect. The normalized absorbance spectrum and probe/pump PL spectra do not share the same y-axis.

Answer: We thank the reviewer for pointing out these important issues regarding Fig. 1a. To improve clarity and visualization, we have adjusted Fig. 1a in the revised manuscript as follows:

The fill color of the probe spectrum was removed to better contextualize the 502 nm and 588 nm absorption peaks. We also labeled these two peaks in the figure.

The y-axes for the normalized absorbance and the probe/pump photoluminescence spectra are now presented separately with clear and correct labels to avoid any confusion.

References Cited in Response Letter:

- [1]. Chachisvilis, M., Kühn, O., Pullerits, T. & Sundström, V. Excitons in photosynthetic purple bacteria: wavelike motion or incoherent hopping? *J. Phys. Chem. B*, 101, 7275–7283 (1997).
- [2]. Billy, J., Josse, V., Zuo, Z. et al. Direct observation of Anderson localization of matter waves in a controlled disorder. *Nature*, 453, 891–894 (2008).
- [3]. Anderson, Philip W. Absence of diffusion in certain random lattices. *Phys. Rev.*, 109, 1492 (1958).
- [4]. Schulze, J., Torbjörnsson, M., Kühn, O. & Pullerits, T. Exciton coupling induces vibronic hyperchromism in light-harvesting complexes. *New J. Phys.*, 16, 045010 (2014).
- [5]. Neutzner, S. et al. Exciton-polaron spectral structures in two-dimensional hybrid lead-halide perovskites. *Phys. Rev. Mater.*, 2, 064605 (2018).
- [6]. Fu, J., Bian, T., Yin, J. et al. Organic and inorganic sublattice coupling in two-dimensional lead halide perovskites. *Nat. Commun.*, 15, 4562 (2024).
- [7]. Park, M. et al. Excited-state vibrational dynamics toward the polaron in methylammonium lead iodide perovskite. *Nat. Commun.*, 9, 2525 (2018).
- [8]. Straus, D. B. et al. Direct observation of electron-phonon coupling and slow vibrational relaxation in organic-inorganic hybrid perovskites. *J. Am. Chem. Soc.*, 138, 13798–13801 (2016).
- [9]. Zhang, H. et al. Ultrafast relaxation of lattice distortion in two-dimensional perovskites. *Nat. Phys.*, 19, 545-550 (2023).
- [10]. Guo, P. et al. Direct observation of bandgap oscillations induced by optical phonons in hybrid lead iodide perovskites. *Adv. Funct. Mater.*, 30, 1907982 (2020).
- [11]. Thouin, F. et al. Phonon coherences reveal the polaronic character of excitons in two-dimensional lead halide perovskites. *Nat. Mater.*, 18, 349-356 (2019).
- [12]. Biswas, S., Zhao, R., Alowa, F. et al. Exciton polaron formation and hot-carrier relaxation in rigid Dion-Jacobson-type two-dimensional perovskites. *Nat. Mater.*, 23, 937–943 (2024).
- [13]. Katan, C., Mercier, N. & Even, J. Quantum and dielectric confinement effects in lower-dimensional hybrid perovskite semiconductors. *Chem. Rev.*, 119, 3140-3192 (2019).
- [14]. Yin, J. et al. Tuning hot carrier cooling dynamics by dielectric confinement in two-dimensional hybrid perovskite crystals. *ACS nano* 13, 12621-12629 (2019).
- [15]. Yin, J. et al. Manipulation of hot carrier cooling dynamics in two-dimensional Dion-Jacobson hybrid perovskites via Rashba band splitting. *Nat. Commun.*, 12, 3995 (2021).
- [16]. Li, X. et al. Two-dimensional Dion-Jacobson hybrid lead iodide perovskites with aromatic diammonium cations. *J. Am. Chem. Soc.*, 141, 12880-12890 (2019).
- [17]. Miyata, K. et al. Large polarons in lead halide perovskites. *Sci. Adv.*, 3, e1701217 (2017).
- [18]. Mauck, C. M. & Tisdale, W. A. Excitons in 2D organic-inorganic halide perovskites. *Trends Chem.*, 1, 380-393 (2019).

- [19]. Lin, R. et al. Dual self-trapped exciton emission with ultrahigh photoluminescence quantum yield in CsCu_2I_3 and $\text{Cs}_3\text{Cu}_2\text{I}_5$ perovskite single crystals. *J. Phys. Chem. C*, 124, 20469-20476 (2020).
- [20]. Li, S., Luo, J., Liu, J. & Tang, J. Self-trapped excitons in all-inorganic halide perovskites: Fundamentals, status, and potential applications. *J. Phys. Chem. Lett.*, 10, 1999-2007 (2019).
- [21]. Shacklette, J. M., & Cundiff, S. T. Role of Excitation-Induced Shift in the Coherent Optical Response of Semiconductors. *Phys. Rev. B*, 66, 045309 (2002).
- [22]. Wu, X., Trinh, M. T. & Zhu, X. Y. Excitonic many-body interactions in two-dimensional lead iodide perovskite quantum wells. *J. Phys. Chem. C*, 119, 14714-14721 (2015).
- [23]. Li, H. et al. The optical signatures of stochastic processes in many-body exciton scattering. *Annu. Rev. Phys. Chem.*, 74, 467-492 (2023).
- [24]. Srimath Kandada, A. R., Li, H., Bittner, E. R. & Silva-Acuña, C. Homogeneous optical line widths in hybrid Ruddlesden–Popper metal halides can only be measured using nonlinear spectroscopy. *J. Phys. Chem. C*, 126, 5378-5387 (2022).
- [25]. Nguyen, X. T., et al. Ultrafast charge carrier relaxation in inorganic halide perovskite single crystals probed by two-dimensional electronic spectroscopy. *J. Phys. Chem. Lett.*, 10, 5414-5421 (2019).
- [26]. Zhang, T., Zhou, C., Feng, X. et al. Regulation of the luminescence mechanism of two-dimensional tin halide perovskites. *Nat. Commun.*, 13, 60 (2022).
- [27]. Narra, S., Lin, C.-Y., Seetharaman, A., Jokar, E. & Diao, E. W.-G. Femtosecond exciton and carrier relaxation dynamics of two-dimensional (2D) and quasi-2D tin perovskites. *J. Phys. Chem. Lett.*, 12, 12292-12299 (2021).
- [28]. Blancon, J.C., Stier, A.V., Tsai, H. et al. Scaling law for excitons in 2D perovskite quantum wells. *Nat Commun.*, 9, 2254 (2018).
- [29]. Dyksik, M. et al. Steric engineering of exciton fine structure in 2D perovskites. *Adv. Energy Mater.*, 15, 2404769 (2025).
- [30]. Lanzetta, L., Marin-Beloqui, J. M., Sanchez-Molina, I., Ding, D. & Haque, S. A. Two-dimensional organic tin halide perovskites with tunable visible emission and their use in light-emitting devices. *ACS Energy Lett.*, 2, 1662-1668 (2017).
- [31]. Jun, T. et al. Lead-free highly efficient blue-emitting $\text{Cs}_3\text{Cu}_2\text{I}_5$ with 0D electronic structure. *Adv. Mater.*, 30, 1804547 (2018).
- [32]. Luo, J., Wang, X., Li, S. et al. Efficient and stable emission of warm-white light from lead-free halide double perovskites. *Nature*, 563, 541–545 (2018).
- [33]. Fang, H.-H. et al. Band-edge exciton fine structure and exciton recombination dynamics in single crystals of layered hybrid perovskites. *Adv. Funct. Mater.*, 30, 1907979 (2020).
- [34]. Dyksik, M. et al. Brightening of dark excitons in 2D perovskites. *Sci. Adv.*, 7, eabk0904 (2021)
- [35]. Blancon, J.-C. et al. Extremely efficient internal exciton dissociation through edge states in layered 2D perovskites. *Science*, 355, 1288-1292 (2017).

- [36]. Mao, L. et al. Hybrid Dion–Jacobson 2D lead iodide perovskites. *J. Am. Chem. Soc.*, 140, 3775–3783 (2018).
- [37]. Stoumpos, C. C. et al. Ruddlesden–Popper hybrid lead iodide perovskite 2D homologous semiconductors. *Chem. Mater.*, 28, 2852–2867 (2016).
- [38]. Yang, Y., Ostrowski, D., France, R. et al. Observation of a hot-phonon bottleneck in lead-iodide perovskites. *Nature Photon.*, 10, 53–59 (2016).
- [39]. Price, M., Butkus, J., Jellicoe, T. et al. Hot-carrier cooling and photoinduced refractive index changes in organic–inorganic lead halide perovskites. *Nat. Commun.*, 6, 8420 (2015).
- [40]. Manser, J., Kamat, P. Band filling with free charge carriers in organometal halide perovskites. *Nature Photon.*, 8, 737–743 (2014).
- [41]. Zheng, K. et al. High excitation intensity opens a new trapping channel in organic–inorganic hybrid perovskite nanoparticles. *ACS Energy Lett.*, 1, 1154–1161 (2016).
- [42]. Baranowski, M. et al. Static and dynamic disorder in triple-cation hybrid perovskites. *J. Phys. Chem. C*, 122, 17473–17480 (2018).
- [43]. Dar, M. I. et al. Origin of unusual bandgap shift and dual emission in organic-inorganic lead halide perovskites. *Sci. Adv.*, 2, e1601156 (2016).
- [44]. Kahmann, S., Shao, S. & Loi, M. A. Cooling, scattering, and recombination—the role of the material quality for the physics of tin halide perovskites. *Adv. Funct. Mater.*, 29, 1902963 (2019).
- [45]. Lee, J., Koteles, E. S. & Vassell, M. O. Luminescence linewidths of excitons in GaAs quantum wells below 150 K. *Phys. Rev. B*, 33, 5512–5516 (1986).
- [46]. Wright, A., Verdi, C., Milot, R. et al. Electron–phonon coupling in hybrid lead halide perovskites. *Nat. Commun.*, 7, 11755 (2016).

We thank the editor and the reviewers for their careful reading of the manuscript (NCOMMS-25-33234A-Z) and for the feedback that has helped us to further strengthen our work. We have revised the manuscript and extended the discussions. We offer answers to reviewers' comments point-by-point and explain the changes made to the manuscript. In the submitted revision, all changes are marked.

Sincerely, on behalf of the authors,

Tõnu Pullerits

REVIEWER COMMENTS

Reviewer #1 (Remarks to the Author):

The authors responded with great detail and additional data and findings to the reviewer comments, which is much appreciated!

This reviewer raised concerns about the novelty, as did other reviewers. It seems the main effects and explanations are mostly expected from what is known from the literature. The response by the authors therefore did not fully rebuttal the concerns about novelty.

Answer: We appreciate the reviewer's continued engagement and the earlier suggestion to consider Anderson localization in interpreting our results. In the revised manuscript, we have incorporated this concept to provide a consistent picture that connects dimensionality, exciton–phonon coupling, and localization phenomena. This integration not only clarifies the mechanism behind the observed stronger coupling in the one-dimensional tin perovskite but also highlights the novelty of our work. To our knowledge, no previous study has pointed out that dimensionality-induced exciton localization can serve as the origin of enhanced exciton–phonon interaction in 1D halide perovskites. This conclusion was possible due to a thorough comparative study of 1D and 2D systems that we are presenting. Thus, by introducing and substantiating this link, our work advances the fundamental understanding of carrier–lattice coupling and disorder effects in low-dimensional hybrid materials. To further emphasise the novelty aspect, we have refined the motivation part of the “Introduction”, see Pages 2-3:

“As reduced dimensionality enhances both quantum confinement and lattice fluctuations, the strength of EPC is inherently sensitive to a material's dimensionality. Although strong EPC has been widely reported in low-dimensional halide perovskites, these effects are usually discussed in terms of enhanced confinement or polaron formation, rather than in connection with dimensionality-induced exciton localization. In particular, the role of such localization in governing EPC remains largely unexplored for organic–inorganic tin halides, where dynamic disorder and soft lattice modes are especially pronounced. Understanding how dimensionality and localization jointly determine EPC provides a new perspective on carrier–lattice

interactions and offers practical routes to tailor the optical and electronic properties of lead-free tin halides for environmentally sustainable applications in solar cells, light-emitting diodes, and coherent light sources.”

We have also refined the “Discussion” part, see Page 15:

“In summary, our combined experimental and computational analysis uncovers the critical role of dimensionality-dependent exciton–phonon interactions in governing both the lattice dynamics and electronic transitions, including exciton self-trapping, in organic-inorganic tin halide systems. This modulation, achieved simply through concentration-controlled tuning of the organic components, provides fundamental insight into the relationship between the structure and functionality in this emerging class of materials. Moreover, it establishes a framework for ligand-mediated control of exciton-phonon coupling, opening new avenues for tailoring the optoelectronic properties of lead-free hybrid metal halides for sustainable photonic and energy applications.”

Regarding previous technical questions, the authors are now claiming that 5ps and 20ps MD provide very similar VDOS, which would be surprising. Comments:

1. can the authors show a comparison of 5ps and 20ps VDOS data?

Answer: The figure is shown below. The results obtained from the PBE functional with and without dispersion correction show similar VDOS spectra.

Fig. R1 The vibrational density of states (VDOS) simulated at (a) 77K, (b) 200K, and (c) 298K.

VDOS is calculated from the velocity autocorrelation function (see below). The later decays rapidly. Even at 77 K, the autocorrelation function has decayed almost to zero at 5 ps (see Fig. S33). This is the main reason why the VDOS from the 5 ps and 20 ps MD runs are so similar.

2. the MD-VDOS data in Figure S33 appear very broad already at low temperature. Why is this the case? Are there experimental data or previous computational data from literature to compare to? Is this behavior expected?

Answer: In classical MD the velocity autocorrelation function (see below) at time zero is:

$$C(0) = \langle v^2 \rangle = 3kT/m .$$

So, the amplitude scales linearly with temperature but the shape of $C(t)$ depends mainly on the dynamics, not directly on T , unless anharmonicity changes the motion. Consequently, also the line shapes do not significantly change with the temperature.

The apparent broadening of the MD-derived VDOS even at 77 K likely arises from anharmonicity of the potentials and soft structure where dynamical disorder, inherent to hybrid perovskites (e.g., molecular cation orientations and local Sn–I distortions), plays a central role. The system has a large number of closely spaced low-frequency modes that merge into a quasi-continuum because of the broadening. For example, in MAPbI₃ broad Raman and THz absorption spectra have been observed even at low temperature (100 K) [Leguy, A. M. A. et al. *Phys. Chem. Chem. Phys.* **18**, 27051–27066 (2016)].

3. the authors should also specify how the VDOS was calculated. If a formula was implemented, the formula should be specified. If other software was used, it should be cited.

Answer: We thank the reviewer for pointing out this omission. We have added the following text with three new references to the Manuscript ('Method', Page 17) [(1) Tang, X. et al., *J. Phys. Chem. Lett.*, **16**, 9656-9663 (2025); (2) Geng, W.-T. et al., *Comput. Phys. Commun.*, **267**, 108033 (2021); (3) Goddard, W. A. et al., *J. Chem. Phys.*, **119**, 11792-11805 (2003)]:

“The vibrational density of states (VDOS) was obtained from the atomic velocity trajectories in the NVT ensemble. For each atom j , the velocity correlation function was evaluated as [70, 71, 72]

$$C(t) = \frac{1}{N} \sum_{j=1}^N \langle \mathbf{v}_j(\tau) \cdot \mathbf{v}_j(\tau + t) \rangle_{\tau}, \quad (1)$$

Where N is the number of atoms and $\mathbf{v}_j(\tau)$ denotes the velocity of the j^{th} atom at time τ .

The VDOS is then obtained from the Fourier transform of VACF,

$$G(\omega) = \int_{-\infty}^{\infty} C(t) e^{-j\omega t} dt . \quad (2)''$$

4. the effect of SOC was found to be around 0.2 eV for the band gap. Is this consistent with previous theoretical work for this or related materials?

Answer: The SOC effect varies depending on both structural orientation and chemical composition of the system. It has been reported to be 0.6 eV and larger in low-dimensional Pb-based perovskites [Pazoki, M., Imani, R., Röckert, A. & Edvinsson, T., *J. Mater. Chem. A* **10**, 20896–20904 (2022)], while a smaller effect is expected in compounds containing Sn. This smaller SOC effect is consistent with the previous theoretical studies on tin halide perovskites such as FASnI₃ and MASnI₃ reporting the values of 0.07-0.40 eV. [(1) Sabino, F. P. et al., *Phys. Rev. B* **110**, 035160 (2024); (2) Peng, L. & Xie, W., *RSC Adv.* **10**, 14679–14688 (2020); (3) Yip, H., Zhao, Y. et al., *J. Mater. Chem. A* **5**, 15124–15129 (2017)].

Reviewer #2 (Remarks to the Author):

Thank you for the detailed responses and thorough revision of the manuscript. Before acceptance, however, I have one main concern that arises from the reply to point 5). The explanation provided for the 502 nm resonance does not appear to be supported by available evidence or theory.

The authors attribute the peak at ~502 nm in the absorption spectrum to “higher-lying excitonic state(s), predominantly 2s but even higher excitons together.” This assignment requires more rigorous justification.

Does for example the energy spacing between the two resonances match the expected 1s–2s energy difference in the 2D sample? More critically, what mechanism could account for a 2s exciton resonance of comparable or even larger intensity than 1s? In a Wannier-like exciton picture, 2s is expected to exhibit a substantially weaker oscillator strength than 1s. In halide perovskites, higher-lying excitons, including 2s, have been hardly observed in optical spectra, even at low temperatures [<https://doi.org/10.1038/s41467-018-04659-x>].

Answer: We agree that, in an ideal Wannier-like exciton model, the 2s exciton would have much weaker oscillator strength than the 1s state. Here, the energy spacing (~ 361 meV) and absorbance ratios between the absorption peaks at ~588 nm and 502 nm in the 2D system are comparable to the reported spectra in layered Sn–I perovskites such as PEA₂SnI₄. [(1) Pitaro, M. et al., *Adv. Mater.* **34**, 2105844 (2022); (2) Yuan, F. et al. *Sci. Adv.* **6**, eabb0253 (2020); (3) Zhang, T. et al., *Nat. Commun.* **13**, 60 (2022)]. In our previous submission, we say: “The 502 nm band is dominated by the 2s transition but most likely includes even higher exciton transitions and the band edge absorption.” So, we do not mean that the whole transition amplitude corresponds to the higher excitons. It may well be that the excitonic contribution is not dominating the absorption; therefore, we have chosen to reformulate the sentence, which now reads: “The 502 nm band originates from the higher exciton transitions together with the band edge absorption.” (see Page 4) This said, we also point out that in these layered

tin-halide perovskites, strong dielectric confinement can modify the exciton wavefunction so that the 1s exciton becomes very localized and screened while the higher excitons are more extended and less screened. This could significantly enhance the relative oscillator strength of the higher excitons compared to the simplest hydrogen-like model. We have added this clarification and the relevant references in Supporting Information (Page 4).

“The energy spacing (~361 meV) and absorbance ratios between the 588 nm and 502 nm absorption peaks in the 2D system agree with reported spectra in layered tin halide perovskites such as PEA₂SnI₄. [17, 18, 19] In the idealized hydrogen-like Wannier exciton model, the 2s and higher excitons are expected to have significantly weaker oscillator strength than 1s. However, strong dielectric confinement in thin layered system can enhance higher-lying excitonic transitions. Therefore, the 502 nm feature is assigned to the higher-lying excitonic resonances combined with the band edge transition.”

Reviewer #3 (Remarks to the Author):

I am pleased with the additional explanations and data sets provided by the authors in this revised manuscript version.

Aside from my initial and still standing remark on somewhat lacking novelty, I am happy to recommend publication of this manuscript in its current form in terms of scientific quality.

Answer: We sincerely thank the reviewer for the positive assessment of the revised manuscript and for recommending it for publication. We also appreciate the continued reflection on the question of novelty. As outlined in our previous responses, the present work introduces a new conceptual link between dimensionality-induced exciton localization and the enhancement of exciton–phonon coupling in tin halide perovskites – an aspect that, to our knowledge, has not been explored before. We are very grateful that the reviewer recognizes the scientific quality of our study and the additional data and explanations we have provided in this revision.

We are grateful to the editor and the reviewer for the thorough evaluation of our manuscript (NCOMMS-25-33234B) and for the helpful comments that enabled us to further refine our study. We have reformulated the equations and carried out the corresponding revisions throughout the manuscript. In the following, we address the reviewer's remarks point by point and describe all implemented changes. All revisions are highlighted in the submitted manuscript.

Sincerely, on behalf of the authors,

Tõnu Pullerits

REVIEWER COMMENTS

All VDOS data in this work were computed with VASPKIT tool using the mass-weighted velocity autocorrelation function ([1] Wang, Vei, et al. *VASPKIT: A user-friendly interface facilitating high-throughput computing and analysis using VASP code. Computer Physics Communications* 267 (2021) 108033; [2] Geng, WT., Liu, YC., Xu, N. et al. *Empowering materials science with VASPKIT: a toolkit for enhanced simulation and analysis. Nat Protoc* 20, 3143–3169 (2025)). In the original manuscript, Equation (1) was written in an oversimplified notation where the explicit mass factors were omitted, which may have caused the confusion; this has been corrected in the revised Methods section. The VDOS spectra discussed in our responses already correspond to the correctly mass-weighted implementation.

Reviewer #1 (Remarks to the Author):

The authors provided answers to my comments. The comment about novelty helped to better understand the relevance of the work.

We thank the reviewer for appreciating the novelty aspects of our manuscript.

The response by the authors to my previous technical comments are not satisfactory:

1. the provided data clearly shows that the VDOS with 5ps trajectory lengths are not converged at all! Textbook knowledge on molecular dynamics provides a rule of thumb that every vibration needs to be sampled at least ten times before it is properly resolved. With vibrational signatures in range of 1 THz or even less in the systems that are studied, this means that the simulation length must be 10 ps or even few times larger. This is seen in the authors' own data, where several new peaks emerge for longer MD runs. If the authors would properly smoothen then 20 ps data, the two curves would not look the same. The authors need to provide further convergence studies (up to 30 or 40 ps, it is the standard in this field) and properly analyze the statistical sampling, in order to finally report technically correct data.

Answer: We thank the reviewer for further emphasizing the importance of convergence in calculating the VDOS. We agree that a short trajectory may artificially broaden low-frequency features. In this work, we employed the ab initio molecular dynamics (AIMD) based on first-principles electronic-structure calculations (including spin-orbit coupling), avoiding the limitations of empirical force fields in describing the strongly anharmonic and dynamically disordered nature of this system. The VDOS spectrum in the previous response is shown with the initial two different AIMD trajectories: an earlier 5 ps trajectory *without dispersion corrections*, and a later 20 ps trajectory *including dispersion corrections*. Significant part of the differences between the results obtained from these two trajectories in our previous answer were due to the different methods. We have now conducted an analysis based on the 20 ps dispersion-corrected trajectory. From this trajectory, we extracted segments of 5, 10, 15 and 20 ps to compute the corresponding VDOS. The results are shown in Fig. R1-2:

- 1) The VACF decays to a value close to zero within 10 ps. After 17 ps it is practically zero.
- 2) While VDOS obtained from all time traces show generally similar features, the difference between VDOS obtained from 5 ps scan and the longer ones is considerable telling that, as the reviewer rightfully pointed out, 5 ps scan is not sufficient for the convergence. Differences between 10 ps, 15 ps and 20 ps scans are minor.
- 3) 77 K VDOS shows more features than 298 K.
- 4) 298 K VDOS is a featureless band starting at about 15 cm^{-1} , has a maximum between 75 to 125 cm^{-1} after which it is slowly reducing in amplitude.
- 5) The 77 K VDOS has a low frequency feature from 20 to 50 cm^{-1} and a dominant sharper band at about 70 cm^{-1} , after that there is a broad feature from 80 to 160 cm^{-1} . Finally, there are two more well-defined features – a weaker one at 180 cm^{-1} and a more pronounced one at around 215 cm^{-1} . Increasing the trajectory length from 10 ps to 20 ps mainly reduces statistical noise and smoothens the shapes; the positions of all major peaks and shoulders remain essentially unchanged. This indicates that, under the simulation conditions (total sampling time of 20 ps), the calculated VDOS is converged within the statistical uncertainty required for the current study.

We would like to emphasize that in this work, the AIMD-based VDOS is used only in a qualitative and supportive manner: (1) to illustrate the temperature dependence of the exciton-phonon interaction, and (2) to confirm that the phonon modes observed experimentally lie in the theoretically expected frequency range. The detailed mode assignment is based on experimental data (oscillation signals in transient absorption measurements) and separate normal-mode calculations, which are already included in the manuscript. Given the size of the system (246 atoms per unit cell, including long-chain organic ligands $[\text{NH}_2(\text{CH}_2)_8\text{NH}_2]$ around the inorganic octahedra) and the use of dispersion-corrected first-principles dynamics, extending the AIMD simulations from 20 ps to 30–40 ps would entail a very substantial additional computational effort, while Fig. R1-2 shows that the main spectral features are already stable within the 20 ps trajectory. We therefore consider the 20

ps dispersion-corrected trajectory to be a reasonable and technically sound compromise for this study. We hope that the reviewer agrees with our argumentation.

Fig. R1 77K. (a) VACF and (b) VDOS of different trajectories: 5 ps, 10 ps, 15 ps, and 20 ps.

Fig. R2 298K. (a) VACF and (b) VDOS of different trajectories: 5 ps, 10 ps, 15 ps, and 20 ps.

2. Figure 2a in the reference provided by the authors shows a Raman spectrum of MAPbI₃ at 100K. The lowest-energy peaks are not broad at all, one can estimate that the FWHM is around 10cm⁻¹ or even less. There is a clear progression of individual peaks, whilst in the current study there are rather broad features, especially at around 100cm⁻¹. Furthermore, the authors' response is insufficient in explaining the origin of it. 77K means only few meV of thermal energy, what evidence is there of anharmonic effects at this low temperature in these materials? Rather, one might hypothesize that the broadening seen is not physical and an error arising from poor sampling: especially at lower T one must take great care regarding statistical sampling and convergence of MD trajectory (see comment 1). One already sees that the noisy and broad 5ps data at a temperature of 77K becomes sharper and more resolved in the 20ps trajectory. This needs to be investigated further, again the data should be technically correct.

Answer: The reviewer is right. We had mixed MAPbI₃ and MAPbCl₃ in our previous response, sorry about that. Indeed, MAPbI₃ has quite narrow low-frequency Raman peaks at 100 K shown in that article. At the same time MAPbCl₃ in the same figure shows a broad band between 20 and 200 cm⁻¹ at 90 K. In the same figure MAPbI₃ too shows very broad featureless Raman spectrum at 125 K. Perhaps even more telling is the THz absorption of MAPbI₃ between 20 and 150 cm⁻¹ with very broad features down to 80 K shown in Fig. 6 of the same article (Barnes, P. R. F., et al., *Phys. Chem. Chem. Phys.* **2016**, *18*, 27051–27066). The broad bands were discussed in terms of “soft semiconductor” by the authors. Inelastic neutron scattering on several hybrid lead-halide perovskites further demonstrates that low-energy optical phonons are weakly dispersive and become overdamped for temperatures as low as ≈80 K, providing direct experimental evidence of strong anharmonicity well below room temperature (Bourges, P. et al., *Commun. Phys.* **2020**, *3*, 48). More recently, Herz and co-workers have highlighted that soft metal–halide perovskites generally exhibit an unusually broad low-frequency Raman response originating from strong anharmonic phonon–phonon coupling and local structural fluctuations (Herz, L. M., et al., *ACS Energy Lett.* **2024**, *9* (8), 4127–4135).

Similar to these studies, we suppose that in this system, the broad feature around 100 cm⁻¹ could be due to a bundle of librational and translational modes of the organic cation [NH₂(CH₂)₈NH₂] that are strongly coupled with tin-iodide vibrations. When finite-temperature anharmonic dynamics are included through AIMD, these closely spaced modes merge into a broad band in the calculated VDOS. On the other hand, as explained in the previous and later response, the calculated VDOS is converged within the statistical uncertainty obtained by Fourier transforming the mass-weighted VACF. Therefore, the broad spectrum is not an artifact of the incorrect methodology or insufficient convergence.

3. Equation 1 provided in the reply is incorrect. C(t) needs to be calculated as mass-weighted velocity autocorrelation function. See Eq. 54 in one of the cited references (Computer Physics Communications 267 (2021) 108033). This is important because without proper mass-weighting the spectrum will be distorted. The authors should implement the correct formula and recalculate all VDOS data.

Answer: We agree with the reviewer that the VDOS must be obtained from the mass-weighted velocity autocorrelation function. In our study, all calculated VDOS data were generated with VASPKIT post-processing tool for the VASP code which uses the mass-weighted velocity autocorrelation function. To remove any possible doubt, we consulted with the corresponding author, Prof. Vei Wang, of the article where VASPKIT is introduced (Computer Physics Communications 267 (2021) 108033), who confirmed that VASPKIT computes the VDOS from the mass-weighted velocity autocorrelation function. Therefore, all VDOS data reported in our manuscript have in fact been generated from the correctly mass-weighted VACF, fully consistent with the reviewer’s comment. We have corrected the equation accordingly.

To avoid ambiguity, we have now revised the Method section accordingly and restored the full formal expression, clearly specifying the mass-weighting step. We have also supplemented the relevant publication that documents the VASPKIT implementation and workflow (*Nat Protoc* **20**, 3143–3169 (2025)). The “Method” part on page 17 is now revised as follows:

“The VACF and VDOS data processing was done by VASPKIT. [70, 71] For each atom i , the velocity autocorrelation function was evaluated as [71, 72, 73]

$$c_i(\tau) = \langle v_i(\tau) \cdot v_i(0) \rangle \quad (1)$$

Where $v_i(\tau)$ denotes the velocity of the i^{th} atom at time τ . The angular brackets denote an average over the entire simulation trajectory. For convenience of comparison, the VACFs are normalized such that $c_i(0) = 1$ in the output. To obtain the overall VACF of the system, the contributions from each atom were combined through a mass-weighted summation:

$$C(\tau) = \sum_{i=1}^N m_i c_i(\tau), \quad (2)$$

Where N is the number of atoms and m_i is its atomic mass. Within the harmonic approximation, the vibrational density of states is then obtained from the Fourier transform of $C(\tau)$:

$$f(\omega) = \frac{1}{k_B T} \int_{-\infty}^{\infty} C(\tau) e^{-i\omega\tau} d\tau \quad (3)$$

where ω is the vibrational angular frequency, k_B is the Boltzmann constant, and T is the absolute temperature. At $\tau = 0$, $C(0)$ is proportional to the total kinetic energy of the system, $C(0) \propto k_B T$, according to the equipartition theorem. The prefactor $1/(k_B T)$ is introduced for normalization purposes and serves to remove the explicit temperature dependence of the mass-weighted VACF. In the present work, we omit this constant prefactor and focus on the spectral shape; therefore, the VDOS is expressed as

$$S(\omega) = \int_{-\infty}^{\infty} C(\tau) e^{-i\omega\tau} d\tau \quad (4).”$$